# Which functional tasks present the largest deficits for patients with total hip arthroplasty before and six months after surgery? A study of the timed up-and-go test phases

Xavier Gasparutto[1]*, Mathieu Gueugnon[2], Davy Laroche[2], Pierre Martz[3,4], Didier Hannouche[5], Stéphane Armand[1]

1 Kinesiology Laboratory, Geneva University Hospitals and University of Geneva, Geneva, Switzerland,
2 INSERM, CIC 1432, Centre for Clinical Investigation, Module Plurithématique, Plateforme d'Investigation Technologique, University Hospital of Dijon-Burgundy, Dijon, France, 3 INSERM, UMR 1093-CAPS, Faculty of Sports, University of Burgundy, Dijon, France, 4 Division of Orthopaedic Surgery and Musculoskeletal Trauma Care, University Hospital of Dijon-Burgundy, Dijon, France, 5 Division of Orthopaedic Surgery and Musculoskeletal Trauma Care, Surgery Department, Geneva University Hospitals and University of Geneva, Geneva, Switzerland

* xavier.gasparutto@hcuge.ch

**Data Availability Statement:** We have uploaded our data on Yareta, an online repository following the FAIR principles. The DOI of the repository is:

## Abstract

Six to eight months after total hip arthroplasty, patients only attain 80% of the functional level of control groups. Understanding which functional tasks are most affected could help reduce this deficit by guiding rehabilitation towards them. The timed up-and-go test bundles multiple tasks together in one test and is a good indicator of a patient's overall level of function. Previously, biomechanical analysis of its phases was used to identify specific functional deficits in pathological populations. To the best of our knowledge, this analysis has never been performed in patients who have undergone total hip arthroplasty. Seventy-one total hip arthroplasty patients performed an instrumented timed up-and-go test in a gait laboratory before and six months after surgery; fifty-two controls performed it only once. Biomechanical features were selected to analyse the test's four phases (sit-to-stand, walking, turning, turn-to-sit) and mean differences between groups were evaluated for each phase. On average, six months after surgery, patients' overall test time rose to 80% of the mean of the control group. The walking phase was revealed as the main deficiency before and after surgery (-41 ± 47% and -22 ± 32% slower, respectively). High standard deviations indicated that variability between patients was high. On average, patients showed improved results in every phase of the timed up-and-go test six months after surgery, but residual deficits in function differed between those phases. This simple test could be appropriate for quantifying patient-specific deficits in function and hence guiding and monitoring post-operative rehabilitation in clinical settings.

10.26037/yareta:vynrqmr3hvegrcnh2oemxkwwb4
https://yareta.unige.ch/#/home/detail/64001c26-
3b96-4978-838a-ea9eaf511f49.

**Funding:** This work was supported by the Division of Orthopaedics and Trauma Surgery at the University Hospitals of Geneva, the "Fondation pour la recherche ostéoarticulaire" of Geneva, the "Programme Hospitalier de Recherche Clinique" and the "conseil régional de Bourgogne Franche-Comté". The funders had no role in study design, data collection and analysis, decision to publish, or preparation of the manuscript.

**Competing interests:** I have read the journal's policy and the authors of this manuscript have the following competing interests: Pierre Martz is a paid consultant for XNOV and SERF. This does not alter our adherence to PLOS ONE policies on sharing data and materials.

## Introduction

Total hip arthroplasty (THA) is a common, cost-effective and usually highly successful surgical procedure to alleviate pain and improve motor function [1] ('motor function' will be referred to as 'function' throughout this paper to improve readability). Although THA improves function for patients with end-stage hip osteoarthritis (OA), patients have not fully recovered function six months and one year after surgery when compared to healthy control groups [2, 3]. On average, patient function increases from 70% of the level of healthy control groups before surgery to 80% six to eight months afterwards [2]. Understanding which domains of function (e.g. strength, balance or mobility) are most improved by THA and which retain the most significant post-surgical deficits could help to map this residual loss of function and guide patient-specific rehabilitation.

Several types of evaluation of function exist. Each covers different domains of functioning as defined by the World Health Organization's International Classification of Functioning and Disability (WHO ICF) [4]. The most common tools are: (i) patient-reported outcomes (PROMs) [5] (e.g. the Hip disability and Osteoarthritis Outcome Score [6]) that evaluate multiple domains of function through self-administered questionnaires (e.g. personal factors, activities, participation); (ii) clinical evaluations that evaluate body function (e.g. hip range of motion); and, (iii) medical imaging that evaluates body structure (e.g. femoral offset). Clinical tests like the 6-minute walk test, the 30-second chair stand test or the Timed Up-and-Go (TUG) test—often described as *performance-based tests* [7]—are used to evaluate patients' capacity as defined by the WHO ICF, i.e. *what a person can do in a standardized evaluation setting* [4].

The TUG test is a robust indicator of a patient's overall level of motor function [8] as it groups together basic mobility skills needed for activities of daily living in a simple, practical test [9]. Participants start sitting in an armchair, stand up, walk three meters, turn back and return to sit in the armchair [9]. The test is recommended by the Osteoarthritis Research Society International [10] for assessing physical function in people diagnosed with hip OA, and it is a noted test of ambulatory activity covering the specific ICF codes of "changing basic body position" (d410), "walking" (d450) and "moving around" (d455), and of strength, agility and dynamic balance [10]. It is widely used to assess mobility in rehabilitation [11].

In the context of THA, pre-surgery TUG test results have been identified as predictive of post-surgery hospital length of stay [12], predictive of ambulatory status six months after a primary THA [13] and associated with a risk of post-surgery deep vein thrombosis [14]. However, as underlined by previous authors [15–17], using the TUG's total time as the sole outcome provides limited information. Indeed, the analysis of the TUG's phase, known as instrumented TUG (iTUG) [18], could provide a deeper understanding of a patient's function before and after treatment. For instance, the iTUG test was shown capable of discriminating between patients with early-to-moderate Parkinson's disease and healthy controls, whereas the total TUG time could not [18, 19]. The iTUG test also showed that slower total TUG times in a group of obese women compared to a control group were explained by the walking and turning phases, not by the sit-to-stand and stand-to-sit phases [17]. To the best of our knowledge, no studies of the TUG test's different phases have been performed previously on patients who have end-stage OA or have undergone a THA. A study to identify the most difficult tasks facing patients before and after surgery could help to target patient-specific rehabilitation.

The present study's objectives were: 1) to assess in which phases of the TUG test patients presented with significant functional deficits compared to a control group, before and after surgery; 2) to assess how these phases changed after surgery; and 3) to compare changes and deficits in total TUG time with changes and deficits in the functional outcomes of the different

phases. To this end, we performed a biomechanical analysis of the TUG test for a group of patients before and six months after surgery, and their outcomes were compared with those of a control group.

## Material & methods

### Participants

This study included 71 THA patients and 52 controls. Participants' characteristics are reported in Table 1. Surgeries were performed by two surgeons using three different approaches: a mini-invasive Rottinger approach (n = 38) [20], a mini-posterior approach (n = 29) [21] or a lateral approach (n = 4), with a dual-mobility cup implanted in all cases. The local ethics committee approved the study (Human Welfare Protection Committee EST I, in Dijon), and it was recorded as a clinical trial (NCT02042586), with all participants giving their written informed consent.

All the patients included in this study were diagnosed with primary hip OA, according to the American College of Rheumatology (ACR) criteria [22], and had stage II, III or IV hip OA according to the Kellgren and Lawrence classification. Mean patient age was 68.5 years old (SD = 8.9 years, range = 46–85 years), with 42 women and 29 men. All had undergone conservative treatment for at least three months before they were considered for a hip replacement, and all were able to understand simple instructions. Exclusion criteria included being younger than 25 years old or older than 85, previous hip surgery, bilateral hip OA, concomitant knee OA, pregnancy or breast-feeding, inflammatory disorders, rapidly progressive hip OA, hip dysplasia, neurological disease, motor neuron disease and all disorders that might interfere with or be worsened by a gait analysis. A control group of participants in the same age range was collected. Their exclusion criteria included OA in the lower limbs, neurological disease, motor neuron disease or disorders that might interfere with or be worsened by a gait analysis.

**Table 1. Participants' characteristics.**

|  | Patients | Control Group | *p*-value |
|---|---|---|---|
| **General Characteristics** |  |  |  |
| Sex (M/F) | 29/42 | 18/34 | 0.607 |
| Age (years) | 68.5 (8.9) | 65.3 (8.6) | 0.054 |
| Height (cm) | 162.8 (8.0) | 163.4 (8.3) | 0.703 |
| Weight (kg) | 75.6 (17.7) | 66.9 (13.0) | 0.001 |
| OA Side (L/R) | (32/39) | - |  |
| **Kellgren and Lawrence classification** |  |  |  |
| II | 14 | - |  |
| III | 31 | - |  |
| IV | 21 | - |  |
| NA | 5 | - |  |
| **Surgical Approach** |  |  |  |
| Rottinger | 38 | - |  |
| Mini-posterior | 29 | - |  |
| Lateral | 4 | - |  |

Sex distribution was tested using the $\chi^2$ test, and differences in age, height and weight used Student's t-tests.

NA = not available.

## Measurement protocol

Participants were equipped with 35 reflective skin markers attached over their whole body according to the Conventional Gait Model [23]. Markers' trajectories were measured at 100 Hz using an eight-camera optoelectronic system (MXT 40, Vicon, Oxford, UK), filtered at 6 Hz using a fourth-order Butterworth design and then occlusions were corrected using marker intercorrelations [24].

Participants were asked to perform the TUG test. They sat in an armchair with a seat at 47 cm off the ground, stood up, walked to a line on the ground three meters away, turned around and came back to sit in the chair [9] (Fig 1). The whole task was made at a self-selected speed, and participants were allowed to use the armrests. Instructions were as follows: "At the start signal, stand up from the chair without using the armrests, if possible, then walk towards the line in front of you, turn back at the line and come back to the chair and sit down, without using the armrests, if possible." Participants performed ten TUG tests, and the fastest of the first three tests was retained for analysis. When sitting, the participant's back could not touch the armchair's backrest because of the motion capture equipment. Patients performed the test before surgery (M0) and six months after surgery (M6), whereas the control group performed the test only once.

## Analysis

The TUG test was divided into four phases (Fig 1): sit-to-stand, walking (back and forth), turn and turn-to-sit [15]. The second turn and stand-to-sit phases were grouped together as a turn-to-sit phase due to the difficulty in discriminating between the end of the turn and the start of the sitting phase [15]. The beginning and end of the TUG test phases were identified using a custom algorithm based on an existing algorithm [25] (S1 File). Work by Beyea et al. [25] was adapted to identify the start and end of turns and the end of the TUG test. Indeed, when checked visually, Beyea's algorithm identified timepoints outside of the phases for those events with the study population. The visual checks of the algorithm's results were assessed using Mokka software [26] and specific plots (S1 and S2 Figs).

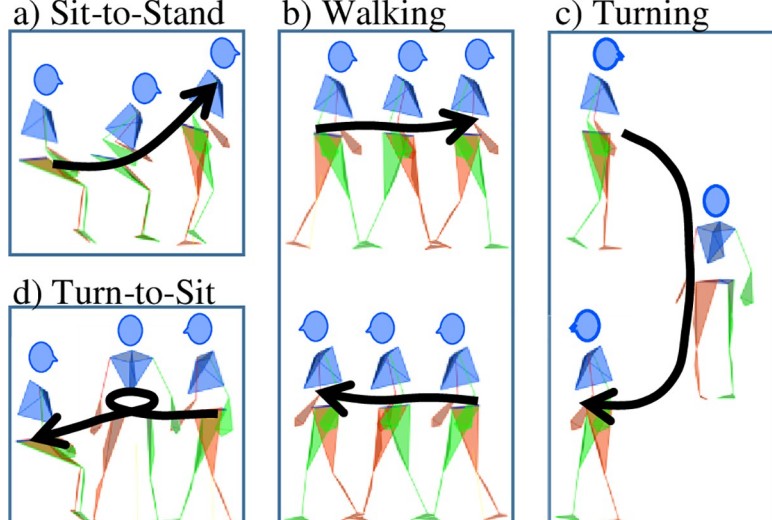

**Fig 1. The four phases of the timed-up and go test.**

The time required to complete each phase and an appropriate set of each phase's biomechanical features were selected for analysis. Feature were selected based on the literature [18, 19, 27–35], and grouped into two categories: 1) quality of movement (joint angles and distances) [36] and 2) speed (linear and angular velocities). We selected a total of 38 features covering the four phases (Table 2) in addition to the total TUG test time and the time for each phase. The reasons for choosing these features and their definitions are detailed in S2 File.

## Statistics

The difference in sex distribution was tested using a $\chi^2$ test. The Kolmogorov–Smirnoff [37] test was used to assess the normality of the age, weight and height distributions among patients and controls. Differences between groups were then tested using unpaired Student's t-tests.

A principal component analysis (PCA) was performed so that only the most relevant biomechanical features were retained for analysis (excluding phase completion times). To account for scale differences, features were centred and reduced by subtracting the mean and dividing by the standard deviation (SD). All three test groups (patients at M0, patients at M6 and the controls) were included in the analysis. The number of principal components was chosen using Cattell's scree test [38]. One feature was selected per principal component and per phase based on the maximal cos2 value and if this value was above 0.20.

The statistical differences between groups for the selected features were assessed using a paired-sample Student's t-test when comparing the patient groups and using an unpaired

**Table 2. Biomechanical features for each phase.**

|  | Quality | Speed |
|---|---|---|
| **Sit-to-Stand** | Peak obliquity thorax | Peak vertical velocity thorax |
|  | Range obliquity thorax | Peak vertical velocity pelvis |
|  | Peak flexion thorax | Peak extension velocity of pathological hip |
|  | Width base of support | Peak extension velocity of contralateral hip |
|  | Length base of support |  |
| **Walking** | RMS obliquity thorax | Peak forward velocity thorax |
|  | Range obliquity thorax | Mean forward velocity thorax |
|  | Lateral RMS C7 | Peak forward velocity pelvis |
|  | Lateral range C7 | Mean forward velocity pelvis |
|  | Range of flex/ext pathological hip |  |
|  | Range of flex/ext contralateral hip |  |
| **Turn** | Number of steps | Peak angular velocity thorax |
|  | Side of turn | Mean angular velocity thorax |
|  |  | Peak angular velocity pelvis |
|  |  | Mean angular velocity pelvis |
| **Turn-to-sit** | Peak obliquity thorax | Peak vertical velocity thorax |
|  | Range obliquity thorax | Peak vertical velocity pelvis |
|  | Peak flexion thorax | Peak extension velocity of pathological hip |
|  | Distance chair to start of turn | Peak extension velocity of contralateral hip |
|  | Number of steps | Peak angular velocity thorax |
|  |  | Mean angular velocity thorax |
|  |  | Peak angular velocity pelvis |
|  |  | Mean angular velocity pelvis |

RMS = root mean square; flex/ext = flexion/extension.

Student's t-test when comparing patient and control groups. The Holm method was used to assess the level of statistical significance (α = 0.05) [39], and Cohen's d was used to assess the effect size. To evaluate the average difference in patient function from control group function, the values of the features at M0 and M6 were reported as percentages of the control group mean. Lower function among patients than among the control group was described as a deficit. Depending on the feature, lower function could be a higher value (e.g. phase duration) or a lower value (e.g. peak vertical velocity of the thorax for the sit-to-stand phase). Changes between M0 and M6 were expressed as percentages of a feature's value at M0 to express post-surgery increases or decreases in function. Furthermore, the percentage of patients with a similar level of function to the control group (i.e. within 1 SD of the control group's mean) was also evaluated, as was the percentage of patients whose function increased between M0 and M6. Pearson correlations were calculated between the total TUG time and selected features for percentage differences to the control group and the percentage of change between M0 and M6. The level of significance was adjusted using the Holm method (α = 0.05) [39]. Only significant correlations were reported.

Finally, the percentages of each feature selected using PCA were averaged for each phase to give one overall figure per phase. Paired Student's t-tests were performed to assess whether the average differences compared to the control group, or between M0 and M6, differed significantly between the TUG test's different phases. To account for multiple testing, the Holm's method was used to adjust the *p*-values (α = 0.05) [39].

## Results

The patient and control groups did not differ significantly in sex, age or height but differed significantly in weight (Table 1).

### Principal component analysis

The complete results of the PCA are presented in S1 Table. Cattell's scree test selected three principal components, representing 56.4% of the total variance. The first principal component (39.1% of variance) was mainly associated with speed parameters, and the second (10.0% of variance) and third (7.3% of variance) with quality parameters.

The biomechanical features selected for each phase are presented in Table 3, and the average scores for those features across the three groups are reported in Fig 2 and Table 4. Each patient's values are compared to the control group's level in S3 Fig.

### Patients' functional deficits compared to the control group

**Deficits before surgery.** The test phase with the largest mean deficit before surgery was the walking phase, at -41% (Table 5). Compared with the other phases, the walking phase deficit was statistically larger than the sit-to-stand, turning and turn-to-sit phases, as per the Holm method. The other phases did not differ significantly from each other. Only 8% of patients had a range of hip flexion–extension during walking similar to the control group's level, whereas 31% to 69% of patients' other features were at the control level (Table 6).

**Residual deficits six months after surgery.** The largest mean residual deficit at M6 was for the walking phase (-22%, 95%CI: -30% to -15%, Table 5). Using the Holm method, this deficit was significantly greater than the sit-to-stand and turning phase deficits, but not greater than the turn-to-sit phase deficit. Only 27% of patients had a range of hip flexion–extension during walking similar to controls, whereas, for other features with significant differences, 46% to 65% of patients had a similar level of function to controls (Table 6).

**Table 3. Features selected using principal component analysis.**

|  | Category | Feature | PC | Cos2 |
|---|---|---|---|---|
| **Sit-to-Stand** | Quality | Peak flexion thorax | 2 | 0.45 |
|  |  | Peak obliquity thorax obliquity | 3 | 0.28 |
|  | Speed | Peak vertical velocity thorax | 1 | 0.61 |
| **Walking** | Quality | Range of flex/ext pathological hip | 1 | 0.50 |
|  |  | Range obliquity thorax | 3 | 0.30 |
|  | Speed | Peak forward velocity pelvis | 1 | 0.86 |
| **Turning** | Quality | Step number | 1 | 0.21 |
|  | Speed | Mean angular velocity pelvis | 1 | 0.66 |
| **Turn-to-Sit** | Quality | Range obliquity thorax | 2 | 0.66 |
|  |  | Distance chair to start of turn | 1 | 0.23 |
|  | Speed | Peak angular velocity thorax | 1 | 0.76 |

PC is the principal component associated with the feature; cos2 ( [0,1]) shows the importance of the feature considered on the associated principal component.

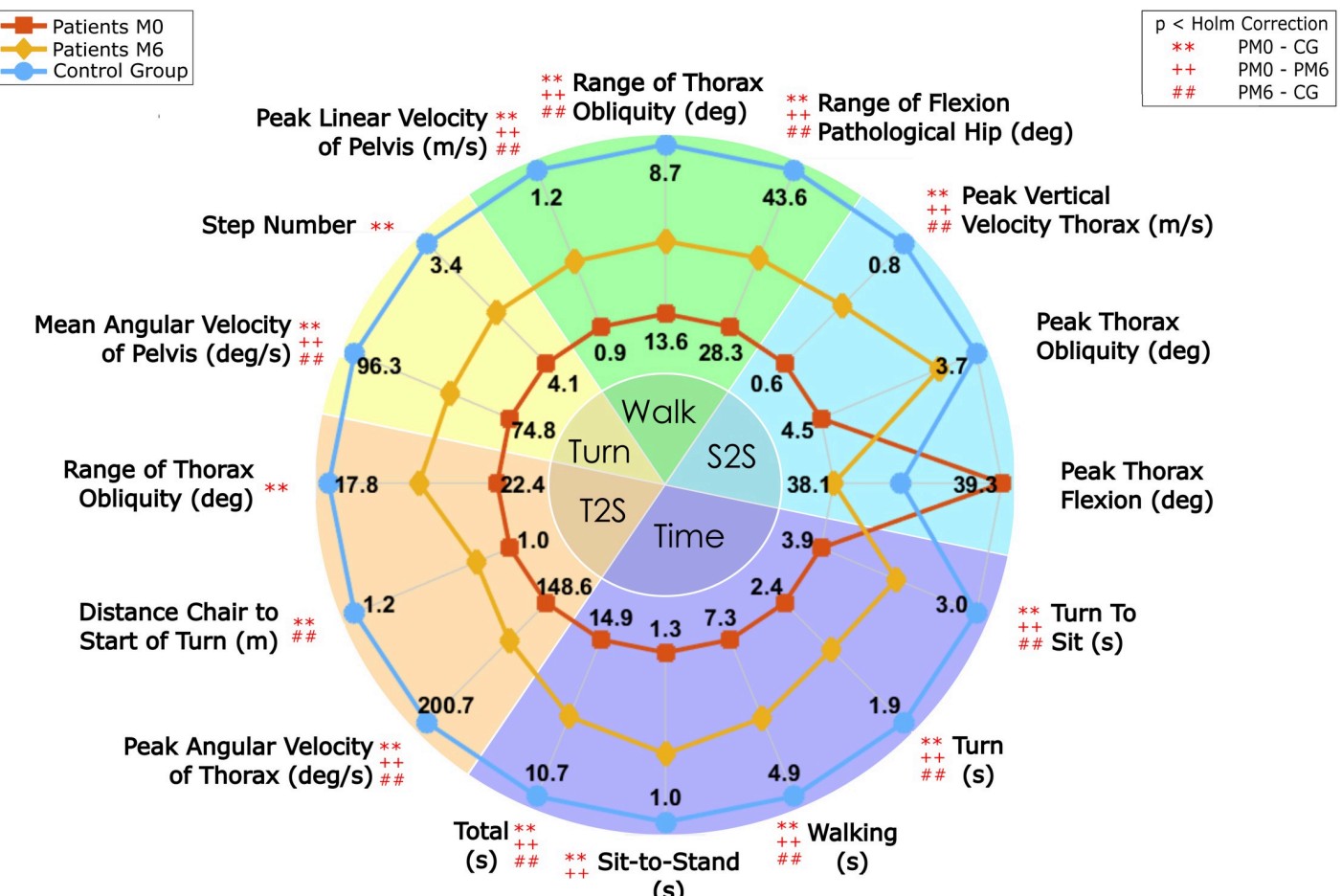

**Fig 2. Overview of each phase's duration and the features selected using PCA for all three groups: Patients before surgery (PM0), patients six months after surgery (PM6) and Control Group (CG).** Feature values representing higher levels of function are on the outer edge, and lower levels of function are on the inner edge. S2S = Sit-to-Stand; T2S = turn-to-sit.

**Table 4. Mean values (standard deviation) for patients before surgery (M0), patients six months after surgery (M6) and the control group.**

| Feature Categories | Features | Patients M0 | | | Patients M6 | | | Control Group | | | Group Comparison | Cohen's d | | |
|---|---|---|---|---|---|---|---|---|---|---|---|---|---|---|
| | | Mean | SD | 95%CI | Mean | SD | 95%CI | Mean | SD | 95%CI | | a | b | c |
| **Total** | | | | | | | | | | | | | | |
| Speed | Time (s) | 14.9 | 4.1 | 13.9 to 15.8 | 12.9 | 2.8 | 12.2 to 13.5 | 10.7 | 2.1 | 10.2 to 11.3 | a, b, c | 1.2 | 0.7 | 0.8 |
| **Sit-to- Stand** | | | | | | | | | | | | | | |
| Quality | Peak thorax flexion (deg) | 39.3 | 10.0 | 37.0 to 41.6 | 38.1 | 8.9 | 36.0 to 40.1 | 38.6 | 10.4 | 35.7 to 41.4 | - | - | - | - |
| | Peak thorax obliquity (deg) | 4.5 | 2.4 | 4.0 to 5.1 | 3.9 | 2.1 | 3.4 to 4.3 | 3.7 | 1.8 | 3.1 to 4.1 | - | - | - | - |
| Speed | Peak vertical velocity thorax (m/s) | 0.56 | 0.19 | 0.52 to 0.60 | 0.66 | 0.17 | 0.62 to 0.70 | 0.77 | 0.17 | 0.72 to 0.82 | a, b, c | 1.1 | 0.6 | 0.6 |
| | Duration (s) | 1.3 | 0.4 | 1.2 to 1.4 | 1.1 | 0.3 | 1.0 to 1.2 | 1.0 | 0.2 | 0.9 to 1.0 | a, b | 0.9 | 0.4 | - |
| **Walk** | | | | | | | | | | | | | | |
| Quality | Range of flex/ext pathological hip (deg) | 28.3 | 7.8 | 26.5 to 30.2 | 35.0 | 7.1 | 33.4 to 36.7 | 43.6 | 5.1 | 42.2 to 45.0 | a, b, c | 2.2 | 0.9 | 1.3 |
| | Range thorax obliquity (deg) | 13.6 | 5.9 | 12.2 to 15.0 | 11.5 | 3.9 | 10.6 to 12.4 | 8.7 | 3.3 | 7.8 to 9.7 | a, b, c | 1.0 | 0.4 | 0.7 |
| Speed | Peak forward velocity pelvis (m/s) | 0.91 | 0.20 | 0.86 to 0.96 | 1.02 | 0.21 | 0.97 to 1.07 | 1.19 | 0.20 | 1.14 to 1.24 | a, b, c | 1.4 | 0.6 | 0.8 |
| | Duration (s) | 7.3 | 2.7 | 6.7 to 7.9 | 6.1 | 1.8 | 5.6 to 6.5 | 4.9 | 1.3 | 4.5 to 5.2 | a, b, c | 1.1 | 0.6 | 0.7 |
| **Turn** | | | | | | | | | | | | | | |
| Quality | Number of steps | 4.1 | 0.9 | 3.9 to 4.3 | 3.9 | 0.9 | 3.6 to 4.1 | 3.4 | 0.9 | 3.2 to 3.7 | a | 0.8 | - | - |
| Speed | Mean angular velocity pelvis (deg/s) | 74.8 | 19.2 | 70.3 to 79.3 | 83.1 | 21.1 | 78.2 to 88.1 | 96.3 | 20.3 | 90.1 to 101.8 | a, b, c | 1.1 | 0.4 | 0.6 |
| | Duration (s) | 2.4 | 0.5 | 2.3 to 2.5 | 2.2 | 0.5 | 2.1 to 2.3 | 1.9 | 0.4 | 1.8 to 2.0 | a, b, c | 1.1 | 0.4 | 0.7 |
| **Turn-to-Sit** | | | | | | | | | | | | | | |
| Quality | Range thorax obliquity (deg) | 22.3 | 9.0 | 20.3 to 24.5 | 20.3 | 7.8 | 18.4 to 22.1 | 17.8 | 6.8 | 16.0 to 19.7 | a | 0.6 | - | - |
| | Distance chair to start turn (m) | 1.03 | 0.20 | 0.98 to 1.08 | 1.06 | 0.17 | 1.02 to 1.10 | 1.17 | 0.16 | 1.13 to 1.21 | a, c | 0.8 | - | 0.7 |
| Speed | Peak angular velocity thorax (deg/s) | 148.6 | 37.6 | 139.9 to 157.4 | 164.9 | 41.9 | 155.2 to 174.7 | 200.7 | 37.2 | 190.6 to 210.8 | a, b, c | 1.4 | 0.6 | 0.9 |
| | Duration (s) | 3.9 | 1.1 | 3.7 to 4.2 | 3.5 | 0.8 | 3.3 to 3.7 | 3.0 | 0.6 | 2.8 to 3.2 | a, b, c | 1.0 | 0.5 | 0.6 |

Significant differences between groups, as per the Holm method, are reported between patients at M0 and the control group as "a", between patients at M0 and patients at M6 as "b", and between patients at M6 and the control group as "c". Cohen's d was reported for significant differences.

## Change in function six months after surgery

The walking phase showed the greatest mean improvement in function (16%, 95%CI: 8% to 24%, Table 5), although this was not statistically significant. The sit-to-stand, turning and turn-to-sit phases showed similar increases (8%, Table 5).

**Table 5. Mean difference ± SD (95% confidence interval) between groups by phase expressed in percentages.**

| Tasks | Patients M0 vs CG | | | Patients M0 vs M6 | | | Patients M6 vs CG | | |
|---|---|---|---|---|---|---|---|---|---|
| | Mean | SD | 95%CI | Mean | SD | 95%CI | Mean | SD | 95%CI |
| Sit-to-Stand (%) | -21 | ± 46 | (-31 to -10) | 8 | ± 48 | (-4 to 19) | -9 | ± 37 | (-17 to 0) |
| Walk (%) | -41 | ± 47 | (-52 to -30) | 16 | ± 35 | (8 to 24) | -22 | ± 32 | (-30 to -15) |
| Turn (%) | -24 | ± 25 | (-29 to -18) | 8 | ± 26 | (2 to 14) | -14 | ± 24 | (-20 to -8) |
| Turn-to-Sit (%) | -27 | ± 38 | (-36 to -19) | 8 | ± 26 | (2 to 14) | -16 | ± 32 | (-23 to -8) |

**Table 6. Mean, SD and 95% confidence interval of the significant differences between groups, i.e. differences between patient and control groups and between patients before and six months after surgery.**

| Feature Categories | Features | Patients M0 vs CG | | | | | Patients M6 vs M0 | | | | | Patients M6 vs CG | | | | |
|---|---|---|---|---|---|---|---|---|---|---|---|---|---|---|---|---|
| | | Mean | SD | 95%CI | % > CG level | R total time | Mean | SD | 95%CI | %Δ >0 | R total time | Mean | SD | 95%CI | % > CG level | R total time |
| **Total** | | | | | | | | | | | | | | | | |
| Speed | Time (%) | -39 | 38 | -48 to -30 | 34 | - | 11 | 17 | 7 to 15 | 72 | - | -20 | 26 | -26 to -14 | 56 | - |
| **Sit to Stand** | | | | | | | | | | | | | | | | |
| Quality | Peak thorax flexion (%) | 2 | 26 | -4 to 8 | 87 | - | 0 | 24 | -6 to 6 | 38 | - | -1 | 23 | -7 to 4 | 86 | - |
| | Peak thorax obliquity (%) | -24 | 65 | -39 to -9 | 75 | - | -5 | 74 | -22 to 12 | 59 | - | -6 | 57 | -19 to 8 | 83 | - |
| Speed | Peak vertical velocity thorax (%) | -27 | 25 | -33 to -21 | 41 | 0.59 | 26 | 37 | 18 to 35 | 73 | 0.63 | -14 | 22 | -19 to -9 | 61 | 0.65 |
| | Time (%) | -33 | 46 | -44 to -22 | 46 | 0.71 | 8 | 33 | 1 to 16 | 70 | 0.57 | -13 | 33 | -21 to -6 | 68 | 0.62 |
| **Walk** | | | | | | | | | | | | | | | | |
| Quality | Range of flex/ext pathological hip (%) | -35 | 18 | -39 to -31 | 8 | 0.37 | 31 | 41 | 22 to 41 | 82 | - | -20 | 16 | -23 to -16 | 27 | 0.40 |
| | Range thorax obliquity (%) | -56 | 67 | -71 to -40 | 49 | - | 5 | 42 | -4 to 15 | 69 | - | -32 | 45 | -42 to -21 | 61 | - |
| Speed | Peak forward velocity pelvis (%) | -24 | 17 | -27 to -20 | 31 | 0.87 | 15 | 22 | 10 to 20 | 79 | 0.77 | -14 | 18 | -18 to -10 | 51 | 0.78 |
| | Time (%) | -49 | 55 | -62 to -36 | 39 | 0.95 | 11 | 26 | 5 to 17 | 73 | 0.83 | -25 | 38 | -33 to -16 | 62 | 0.92 |
| **Turn** | | | | | | | | | | | | | | | | |
| Quality | Number of steps (%) | -20 | 26 | -26 to -14 | 69 | - | 4 | 28 | -3 to 10 | 42 | - | -12 | 27 | -18 to -5 | 77 | 0.45 |
| Speed | Mean angular velocity pelvis (%) | -22 | 20 | -27 to -18 | 41 | 0.61 | 14 | 28 | 8 to 21 | 68 | 0.52 | -14 | 22 | -19 to -9 | 61 | 0.64 |
| | Time (%) | -28 | 28 | -35 to -21 | 41 | 0.68 | 6 | 23 | 1 to 11 | 66 | 0.49 | -17 | 26 | -23 to -11 | 58 | 0.68 |
| **Turn to Sit** | | | | | | | | | | | | | | | | |
| Quality | Range thorax obliquity (%) | -25 | 51 | -37 to -14 | 69 | - | 3 | 31 | -4 to 11 | 52 | - | -14 | 44 | -24 to -3 | 76 | - |
| | Distance chair to start turn (%) | -12 | 17 | -16 to -8 | 51 | 0.49 | 6 | 22 | 1 to 11 | 61 | - | -9 | 15 | -13 to -6 | 63 | 0.45 |
| Speed | Peak angular velocity thorax (%) | -26 | 19 | -30 to -22 | 31 | 0.73 | 13 | 23 | 8 to 18 | 72 | 0.67 | -18 | 21 | -23 to -13 | 46 | 0.78 |
| | Time (%) | -31 | 37 | -39 to -22 | 44 | 0.76 | 8 | 21 | 3 to 13 | 68 | 0.71 | -16 | 27 | -22 to -9 | 65 | 0.76 |

"% > CG" represents the number of patients with function similar to the control group's level (within 1 SD of the mean) for that feature, and "% Δ > 0" represents the number of patients with an improvement in that feature's function at M6. "R total time" represents the feature's coefficient of correlation with total test time. Features with no significant differences between the groups considered are reported with light grey background.

## Total TUG times

Mean total TUG times were 14.9 ± 4.1 s and 12.9 ± 2.8 s for patients at M0 and M6, respectively, and 10.7 ± 2.1 s for the control group (Table 4). On average, patients were -39% (95% CI: -48% to -30%) slower than the control group before surgery, and they had improved their total time by 11% (95%CI: 7% to 15%) six months after surgery. However, they were still -20% (95%CI: -26% to -14%) slower than the control group (Table 6). These differences were all

statistically significant. The majority of patients (72%) improved their total time between M0 and M6. At M0, 34% of patients had a functional level similar to the control group, rising to 56% at M6 (Table 6).

**Associations between deficits and changes in features and deficits and changes in total TUG times (Table 6).** The deficits and changes in total TUG time were mainly associated with speed parameters, especially with the deficits and changes in the duration of the walking phase and with the peak forward velocity of the pelvis (Table 6). The deficits in the range of flexion–extension of the patient's pathological hip during walking at M0 and M6, the deficit in the number of steps taken during the turning phase at M6, and the deficits in the distance from the chair to the start of the turn at M0 and M6 were the only quality parameters associated with the difference in total time that displayed moderate correlations (Table 6). The change in total TUG time between M0 and M6 was only associated with speed parameters. The highest correlations were with the change in the time of the walking phase and the change in peak walking speed.

## Discussion

The present study assessed the change in function among a group of patients with THA, before and six months after surgery, and the functional deficits in comparison to a control group by the means of an instrumented TUG test. The test was divided into four phases [15], representing different activities of daily living (sit-to-stand, walking, turning and turn-to-sit); multiple biomechanical features quantifying the motor functions were selected using PCA and, the features for each phase were compared between groups. Finally, mean pre- to post-surgery changes and deficits in features, phase times and phase functional outcomes—between patients at those times and compared with the control group—were assessed to evaluate links between them and deficits and changes in total TUG time.

### Patients' functional deficits compared to the control group

Results suggested that the walking task represented the main limitation to patients with THA both before and six months after their surgery. The changes in walking revealed in this study were consistent with a recent literature review showing that six months post-surgery, gait speed and the pathological hip's range of flexion–extension increased, although they remained lower than control group levels [3]. Our study's results showed that the hip range of flexion–extension during walking seemed to be an appropriate feature of motor function with which to study the effects of THA. Indeed, this feature presented the lowest percentage of patients with a level of function similar to that of the control group, both before and after surgery (8% and 27%, respectively), as well as the greatest number of patients with a positive change in function after surgery (82%). This suggests that, despite their deficits six months after surgery, THA has a positive effect on patients' sagittal hip kinematics during walking.

### Changes in function six months after surgery

There were no significant differences in the changes between the phases, although the walking phase showed the largest mean increase in function. Thus, THA seems to improve patients' motor functions homogeneously across the different phases of the TUG test.

### Changes and deficits in total TUG time compared to changes and deficits in functional outcomes

The changes and deficits in total TUG time were mainly strongly to moderately correlated with the changes and deficits in the test's speed features, but also with a few features

representing the quality of movements (pathological hip ranges of motion at M0 and M6, number of steps during the turn at M6, and distances from the chair to the start of turn at M0 and M6). More specifically, the changes and deficits in total test time were strongly correlated with the changes and deficits in the duration of the walking phase, the TUG test's longest phase. Total TUG test time could be a good indicator of speed parameters, but it does not seem to reflect comparably the quality of movements, such as the presence of compensatory movements (e.g. high range of thorax obliquity). Moreover, Caronni et al. [15] showed that an improvement in total TUG time was not associated with improvements in all the test's phases. Thus, this feature's utility seems limited in improving our understanding of a patient's level of function across multiple domains.

Nevertheless, the residual deficit in total TUG time was 20% compared to the control group mean. This is consistent with the 20% residual deficit in function reported by Vissers et al. six to eight months post-surgery [2], and it supports a previous study suggesting that the total time of the TUG test could be a good indicator of the overall level of motor function [8].

## Outcome variability

Although THA patients' function improved in every phase of the TUG test after their surgery, on average, they nevertheless presented with deficits in each one of those phases when compared to the control group before and six months after their surgery. This supports previous studies showing improvements in function but an incomplete recovery six months after surgery [2]. However, the SDs of the percentage differences between groups were very high, e.g. the SD of the deficits in total TUG time at M0 was 38% and the mean deficit was 39%. These high SDs suggest great variability in patient profiles. Indeed, at M6, some patients had similar levels of function to the control group, whereas others showed no improvements in function (S3 Fig). Understanding the preoperative factors at the origin of those differences in profile could help to identify the patients at risk of having a lower level of function after surgery and improve patient rehabilitation.

The high variability resulting from the analysis of the TUG test's phases suggests that it could be used to identify patient-specific levels of function. Identifying a functional deficit in a particular phase might help target the functional domain in which the patient is most limited. For example, a patient with a large post-surgical deficit in the sit-to-stand phase could be oriented towards specific strength training to reduce this limitation [33]. Such tests could also be used to monitor patients' rehabilitation processes post-surgery.

The PCA showed that the largest part of the variance between groups was associated with speed features. This seems to indicate that patients improve the speed of their movements more than they improve their quality at six months after surgery. One explanation could be that the reduction in post-surgery pain leads to improved hip function and thus to increases in speed, but that the compensations developed before surgery, to avoid pain in the arthritic hip, might take more than six months to correct. Nevertheless, the range of thorax obliquity was significantly lower after surgery for the test's walking and turn-to-sit phases, which indicates a decrease in compensatory movements during walking and less asymmetry during the sitting phase.

Surprisingly, the direction of the turn was not a relevant parameter for this pathology. Indeed, at M0 and M6, half the patients turned towards their pathological side and the other half towards their non-pathological side.

## Implementation in clinical settings

The TUG test could be performed with a simpler, low-cost setup by using Inertial Measurement Units (IMUs), as previously done with other pathological populations [18]. Only three

IMUs placed on the thorax, pelvis and thigh of the pathological hip would be necessary to measure all the features used in the present study. The iTUG test could be a simple and cost-effective means of assessing the functional deficits of patients with hip OA before and after THA, and it could help healthcare professionals to target patient-specific rehabilitation to close the gaps in function with asymptomatic controls. As recommended by the COSMIN taxonomy, the duration of the phases and the features should be assessed for reliability, validity and responsiveness before they are translated to clinical practice [40]. Recently, and with encouraging results, responsiveness to changes in multiple iTUG test parameters and the minimal detectable change in total TUG time have been evaluated among older adults (63.9 ± 6.1 years) undergoing a six-week physiotherapy programme [41]. Indeed, responsiveness to change was shown to have small to moderate effect sizes, and the minimal detectable change in total TUG time was 0.77 s. Moreover, Smith et al. [41] obtained positive feedback from patients and clinicians when implementing such an iTUG test in clinical settings, suggesting its good feasibility and acceptability. Nevertheless, further studies are needed to identify whether using such tests before surgery could help predict patients' post-surgical function.

## Limitations

The present study's main limitation is its lack of one-year and ongoing follow-up to assess changes in patient function in the long term, which is key to understanding the success of THA [42].

The patient population and the control group had a significant difference in terms of weight, but greater weight has been reported as a risk factor for hip OA [43], which could explain this.

The duration of test phases and the time differences between groups were relatively small, especially for the sit-to-stand and turning phases. Thus, those results should be taken with care as the difference could be within measurement accuracy of the algorithm. Moreover, data centring and reduction led to the selection of parameters with non-clinically meaningful differences. Indeed, the PCA identified parameters with differences that were within the range of measurement accuracy (1 degree), e.g. the peak thorax flexion and range of thorax obliquity during the sit-to-stand phase. Future studies could use measurement accuracy as a criterion during the feature selection.

The features measured in the present study were chosen based on previous ones, but other features might also be relevant when assessing levels of function in each phase. For example, we observed visually that patients moved within a continuum between two strategies in the turn-to-sit phase: those who seemed the most impaired walked right up to the chair, turned and then sat down, and those who seemed less impaired could turn and sit in one continuous movement, including one or more backward steps. These two strategies were similar to those described by Weiss et al. [35] among elderly patients: the distinct transition strategy and the overlapping transition strategy. The "distance chair to start of turn" feature was chosen with the assumption that a longer distance would imply that patients turned and sat simultaneously, i.e. using an overlapping transition strategy. This feature was continuous, as this study aimed to assess percentages of differences and change, but a categorical feature classifying patients' strategies could be of interest to understand patients' motor function.

## Conclusion

This study showed that, on average, patients with THA presented deficits in all four phases of the TUG test when compared to a control group before and six months after surgery, even though their motor function improved in all those phases after surgery. Walking seemed to be

the main limitation facing patients before and six months after surgery. Furthermore, the large variability in functional level observed between patients indicates that the iTUG test was able to capture that variety among this population. Thus, this test could be appropriate for evaluating patient-specific levels of function in clinical settings. A biomechanical analysis of the TUG test with a simple IMU-based measurement system could provide cost-effective, relevant standard information for identifying patients' specific functional deficits and could help target and monitor patient-specific rehabilitation in clinical practice.

## Supporting information

**S1 Fig. Figures to check the phases of patients.**
(PDF)

**S2 Fig. Figures to check the phases of the control group.**
(PDF)

**S3 Fig. Comparison of patient's features and times of phases with the level of the control group.**
(PDF)

**S1 Table. Results of the principal component analysis.**
(PDF)

**S1 File. Algorithm for the detection of the timed up-and-go phases.**
(PDF)

**S2 File. Features description and selection.**
(PDF)

## Author Contributions

**Conceptualization:** Xavier Gasparutto, Davy Laroche, Pierre Martz, Stéphane Armand.

**Data curation:** Mathieu Gueugnon, Davy Laroche, Pierre Martz.

**Formal analysis:** Xavier Gasparutto, Stéphane Armand.

**Funding acquisition:** Didier Hannouche.

**Investigation:** Xavier Gasparutto, Stéphane Armand.

**Methodology:** Xavier Gasparutto, Mathieu Gueugnon, Davy Laroche.

**Software:** Xavier Gasparutto, Mathieu Gueugnon.

**Supervision:** Didier Hannouche, Stéphane Armand.

**Writing – original draft:** Xavier Gasparutto, Stéphane Armand.

**Writing – review & editing:** Xavier Gasparutto, Mathieu Gueugnon, Davy Laroche, Pierre Martz, Didier Hannouche, Stéphane Armand.

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
