## [Decision Letter · Decision Letter 0]

11 Feb 2021

PONE-D-20-38214

Which functional tasks present the largest deficits for patients with total hip arthroplasty before and 6 months after surgery? A study of the Timed Up-and-Go phases

PLOS ONE

Dear Dr. Gasparutto,

Thank you for submitting your manuscript to PLOS ONE. After careful consideration, we feel that it has merit but does not fully meet PLOS ONE’s publication criteria as it currently stands. Therefore, we invite you to submit a revised version of the manuscript that addresses the points raised during the review process.

Specifically, both reviewers requested more details on several items in the methods section of the paper.

We look forward to receiving your revised manuscript.

Kind regards,

Peter Andreas Federolf

Academic Editor

PLOS ONE

Journal Requirements:

3.Thank you for stating the following in the Competing Interests section:

"I have read the journal's policy and the authors of this manuscript have the following competing interests: Pierre Martz is a paid consultant for XNOV and SERF."

Reviewers' comments:

Reviewer's Responses to Questions

**Comments to the Author**

1. Is the manuscript technically sound, and do the data support the conclusions?

Reviewer #1: Partly

Reviewer #2: Partly

2. Has the statistical analysis been performed appropriately and rigorously? 

Reviewer #1: Yes

Reviewer #2: No

3. Have the authors made all data underlying the findings in their manuscript fully available?

Reviewer #1: Yes

Reviewer #2: Yes

4. Is the manuscript presented in an intelligible fashion and written in standard English?

Reviewer #1: Yes

Reviewer #2: No

5. Review Comments to the Author

Reviewer #1: General

The study investigated changes in an instrumented TUG test in patients before and 6 months after total hip replacement and compared them to healthy controls. They predefined biomechanical features according to the literature and used a PCA analysis to reduce the number of features. The selected features were then compared between groups. However, the description of the biomechanical features and especially the statistical analysis section is insufficient and makes it difficult completely follow and understand the analysis and results. If clarified, the paper presents interesting and clinically relevant data of a simple test that could be used in clinical practice to assess the functional outcome after total hip replacement.

Major comments:

Analysis/Table 2: The biomechanical features are insufficiently explained. Abbreviations are not explained (table 2). How was angular velocity of thorax/pelvis defined? In a plane/resultant? With respect to what point/segment? How was side of turn defined? How was pathological/contralateral defined in controls?

Statistics

How were the biomechanical features reduced?

You state that you kept one feature per component and per subtask. What do you mean by component? PC component? Why do you keep Peak angular velocity thorax and peak vertical velocity thorax in turn to sit which are both associated with PC1 (table 3)?

Bonferroni correction: what is the number of tests?

It is unclear why calculate the percentage of variation. From my understanding of the results, this is not presented anywhere and not related to the aims of the study. Please clarify.

Results

You often write about a deficit. But it is not explained in the methods how this deficit was defined.

Numbers in tables, figures and text are repeatedly different from each other even though they should be same (i.e. Fig.2 and Table 4 values for control group).

Why do you use different thresholds for presenting p-values (i.e. Table 4 and Fig 2. once p<0.001, once p<0.003)?

You write about a statistically larger deficit in some phases. It is difficult to understand how you derive these results. It would be helpful to directly show the results of the statistical comparison between subtasks (incl p-values and corresponding threshold of Bonferroni correction)

Figure 2: The figure caption states that Fig 2 presents an overview about the features selected by the PCA. However, it also contains the duration of the different TUG phases. From my understanding, these were not included in the PCA. Please clarify. Moreover, an explanation on how the points in the plot were selected is missing (sometimes larger number is inside, sometimes outside. How was position of point for M6 defined?).

Minor comments:

Abstract

Line 38: Do you mean “improve”? What parameter? Shorter duration of the phases?

Line 39-41: You conclude that the iTUG could be instrumented with a wearable sensor in a clinical setting. However, from your paper, it’s not clear how that would be achieved. Please revise.

Introduction

Line 47/48: This should be “6 months and 1 year after THA”

Lines 61-62: Please write “test” after each test (6minute walk test, 30s chair stand test or TUG test), or only after TUG test.

Lines 78ff: Please specify what parameters are measured/analyzed in the iTUG.

Methods

Lines 107-109: You state that all patients had either stage II, III or IV hip OA, but in Table 1 K/L classification is missing for 4 patients. Why are these data missing?

Line 118: Please change to “destabilised” (you used British English spelling in the rest of the manuscript).

Line 119-120: Please change to “between groups” (or “between populations”)

Line 131: There is a space missing between “chair” and “without”

Lines 133ff: Why did you perform 10 TUGs but only analyze the fastest trial of the first 3 TUGs? Please explain.

Line 147/S1: Is there any information on what S1 shows? The figure is not self-explanatory, it’s not clear what the sticks indicate.

Line 175: Please change to “one global figure per phase”.

Line 176: I think you mean “development” instead of “evolution”. Please change throughout the manuscript (evolution and evolve).

Results (there are no line numbers after page 14, so comments were made on paragraphs)

Table 4: Please check that you use the same precision for mean and SD. Was the SD of mean angular velocity pelvis (turn) and peak angular velocity thorax (turn to sit) really below 1?

Line 206: Table 6 appears before Table 5 in the text. Please change

Line 209: This should be “Bonferroni correction”

Table 5: What do bold numbers indicate? What is parameter corr. total time?

Table 6: Please specify in legend, what parameter is meant by percentage.

Table 5/6: Were the 95% CI of the difference really that much smaller than the SD of the difference?

P.14, paragraph 1, last sentence (49-86%): why do you here include all parameters and in previous paragraph only those with significant differences?

Discussion

P.15, paragraph 2: what scores of the patients? This implies to me that you compared PC scores, but from the results it’s not what you did. Please clarify.

P.15, paragraph 3, last sentence: You state that this (hip flex/ext range walking phase) has a strong positive effect. However, only 20% of patients reach level of controls. Isn’t this contradictory?

P.17, variability of outcomes: From the presented results, I don’t understand how you analysed variability.

P.17, paragraph 1: were there any patients that had decreased performance 6months postoperative?

P.17, paragraph 2: please change “strong post-surgery deficit” to “large post-surgery deficit”

P.18, paragraph 1: Please change to “three synchronized IMUs…”

P.18, paragraph 2: you state that a simple 1 IMU/smartphone provides enough parameters to assess the function of the patients. However, from your analysis this is not valid. The presented parameters were not measured with IMUs and would need several sensors (data on pelvis, hip, thorax…). Please elaborate on how such a sensor should be set up according to your results.

P.18, paragraph 2, last 4 lines: Please use either “a test” or “tests”. Please change to “However, further studies are needed…”

P.18, paragraph 4. Please change to “duration of the phases” instead of “time of the phases”.

P.19, paragraph 1: You state that there was a study on minimal detectable changes of iTUG parameters. How do the difference between pre and post patients compare to these minimal detectable changes? Please elaborate.

P.20, paragraph 1: You abbreviated timed-up and go test as TUG test, please change.

Reviewer #2: The study explores functional deficits before and its change after total hip arthroplasty with an instrumented timed up-and-go test, analyzing phase specific differences between patients and a control group. The topic is of clinical relevance and the sample size adequate to produce meaningful results. The relevance and experimental setup are well described and introduced. Limitations are within the study design, where no post-measurement of the control group is available, resulting in multiple testing and a reduction in statistical power. Further methodological shortcomings, including the variable selection, reduce the significance of the results. In some cases, false or non-ideal statistical tests are used and in other cases (e.g. PCA), methods are not well enough described to judge the correctness of the results. Combined with further language and typing errors, major revisions are necessary to improve the manuscript. Nevertheless, the study has the potential to come to good and meaningful results.

To improve the study/manuscript following main issues should be revised:

1. The preselection of variables is arbitrary and not theory or hypothesis driven. The sole explanation are previous studies, who investigated other disorders and included further variables which have not been considered in the current study (e.g. spatiotemporal variables, such as cadence, stride length, length of single support…). Ideally the variables should be based on known limitations in hip OA patients, identified in previous studies. Furthermore, the joint angles over the complete range of motion (ROM) are unspecific, making conclusions about deficits for specific movements or walking phases difficult. For example, dividing hip ROM in flexion and extension would help to interpret whether deficits during walking are more related to the late stance phase (restricted hip extension), indicating problems to generating step length or swing phase (restricted hip flexion), which might result in higher fall risk due to lower foot clearance. It is unclear why peak thorax obliquity was only analyzed during sit2stand/turn2sit and not walking/turning and why peak thorax obliquity was not directly investigated of the pathological side? Conversely, what is the rationale behind investigating different mean velocities of the pelvis and thorax. In which scenario do the authors expect to find notable differences in the mean velocities of these segments?

2. PCA:

The PCA might be a good approach to identify important variables. However, insufficient clarity and scarce description make it impossible to judge the results. Especially in the presence of differently scaled variables the standardization and re-scaling is of utmost importance for reliable results. It is unclear how the variables were centered and what is meant by “reduced”. Please clarify.

It is not described if both groups were included in the PCA analysis.

Why was the time not included? Wouldn’t that give important information which quality features are related to the time? Doing so might also spare the correlations, which further complicate the analysis.

The choice of variables based on the cos² values is inconsistent. For example, in quality during walking “Range of flexion contralateral hip (0.47)” and “Range obliquity thorax (0.30)” have higher cos² value than “RMS obliquity thorax (0.27)”. The selection method or the description needs clarification.

3. The Kolmogorov-Smirnoff test is inappropriate to compare the patient characteristics. Age distribution can be the same in a group of children and adults. The interesting information is, whether the patient and control group are of similar age. Therefore, t-tests are needed, and the K-S tests only serves to test the assumption of normal distribution for the t-tests. For weight, a non-parametric test like the Mann-Whitney U test is necessary.

4. Number of tests and p-value correction:

The fact that the study design does not allow for repeated measures of variance and the downside of multiple testing is a problem. The current approach with the Bonferroni method is very conservative: if I count right 16*3 t-test were calculated for the variables plus 4 for the phases, resulting in a significance level of 0.05/52 <= 0.001, which conversely inflates the false negative (type II error) risk.

I would recommend controlling the FDR with the Benjamini-Hochberg correction or the Holm method.

To further strengthen the study effect sizes with CI could be calculated.

5. Overall deficit in percent:

The approach to calculate the overall deficit only on the significantly different variables within a phase is questionable. What if there was no significant variable within a phase? Has this phase gets no overall rating? What about cases where significant differences exist only pre- or post-surgery? Is the variable only included in one of the overall scores but not in the other?

Overall score should include all important variables (already identified by the PCA) and differences tested afterwards.

Further minor revisions:

Note: These corrections are not exhaustive since it can be expected that major changes to the manuscript will follow the main revisions. Therefore, the corrections are rather examples hinting towards deficits requiring attention.

1. Further remarks:

• “52 aged-matched controls” if they were age-matched, there would not be a 3y difference in age.

• “pregnancy or breast-feeding patients” relevant in this age group?

• “The patients and control groups did not differ in sex, […]” as pointed out earlier, the K-S test does only test distribution differences and gives no insight about mean group differences.

• Table 4: Please revise/double-check. SD of Mean angular velocity pelvis (deg/s) (turn) and Peak angular velocity thorax (deg/s) (turn2sit) at M6 appear wrong.

Use the same number of digits for a variable mean and SD.

• Table 5 is inconsistent. Sometime coloring is false (e.g. SD from Range thorax obliquity (%) during walking pre-post comparison). Also reporting of correlations is inconsistent (sometimes left out despite sig. diff and sometimes displayed without sig. diff.)

• “This deficit was statistically larger than the other phases (p < 0.05) but not with the Bonferonni correction.“ Reporting „significant“ results, which are insignificant after p-value correction goes against the logic of doing a correction.

• “[…] selected by PCA and the scores of the patients […]” implies that PCA scores were compared, which is not the case.

• “[…] presence of a limp (indicated by the range of thorax obliquity).” Limping might not be very well represented by the thorax ROM. It does not reveal if the trunk lean was one-sided. Furthermore, thorax ROM might not be a good measure of limping because there are probably ways of limping that don’t involve an excessive trunk lean. Letter is a good strategy to unload the hip and is a known compensatory movement for weak hip abductors or hip pain. For limping, however, a better and more direct measure would be the single support phase.

• The section “Implementation in clinical settings” might be well-meant, however, has too little relation to the results.

• “The reliability, validity and responsiveness should be assessed for the time of the tasks and for the features of this study before using them in clinical practice, […]” Limitations should relate to the study.

• Last paragraph of Limitations: “Surprisingly, the PCA identified parameters…” is not a limitation and belongs in the discussion.

2. Variable names:

The variables should be named more precisely.

“Lateral distance between feet”. Is that step width? There is no lateral distance between the feet.

“Only 8% of patients had a range of hip flexion extension during walking […]” variable name in tables implies only hip flexion was analyzed. If the variable comprises hip flexion/extension it should be named accordingly.

3. Language and typos

Language and wording are not quite yet on a publication level and should get revised by a native speaker or expert.

Examples for awkward phrasing, wording, grammar, and typos:

• “Exclusion criteria included a previous surgery to the hip” � previous hip surgery.

• “hospital length of stay after surgery” � hospitalization?

• “interfere or be destabilised by gait analysis” � interfere with or be worsened by …

• “(i.e. above the mean of the control group minus one standard deviation)” � within 1SD of the control group

• “Evolution” is the wrong word to describe the change/difference between pre/post

• “pace” unusual in this context. Speed is probably more appropriate.

• “TUG test was not performed previously for patients with end stage” � previously in patients.

• “up-an-go” � up-and-go.

In conclusion, there is much room for improvement and besides the major issues all the little errors throughout the manuscript and tables indicate that the manuscript is not ready for publication. Nevertheless, the study might provide very useful information to the scientific community.

6. PLOS authors have the option to publish the peer review history of their article (what does this mean?). If published, this will include your full peer review and any attached files.

Reviewer #1: No

Reviewer #2: No

---

## [Author Response · Author response to Decision Letter 0]

30 Mar 2021

Dear editor and reviewers,

We would like to thank the reviewers for their thorough analysis of our paper and the highly relevant observations made on the method, presentation of results, and points to clarify. These comments helped us increase significantly the quality of the paper as well as clarifying the outcome of the study.

The following document gives a detailed answer of the comments of each reviewers. Our answers are written after each comments of the reviewers.

We hope that these answer will satisfy the reviewers and we remain at your disposal for any additional clarifications,

Best regards,

The authors

 

Reviewer #1: 

General

The study investigated changes in an instrumented TUG test in patients before and 6 months after total hip replacement and compared them to healthy controls. They predefined biomechanical features according to the literature and used a PCA analysis to reduce the number of features. The selected features were then compared between groups. However, the description of the biomechanical features and especially the statistical analysis section is insufficient and makes it difficult completely follow and understand the analysis and results. If clarified, the paper presents interesting and clinically relevant data of a simple test that could be used in clinical practice to assess the functional outcome after total hip replacement.

- The statistical analysis and its description was modified based on the comments of reviewer 1 and 2 and we added a supporting information file (S4) to give details on the computation of the biomechanical features and their selection.

Major comments:

Analysis/Table 2: The biomechanical features are insufficiently explained. Abbreviations are not explained (table 2). How was angular velocity of thorax/pelvis defined? In a plane/resultant? With respect to what point/segment? How was side of turn defined? How was pathological/contralateral defined in controls?

- A supplementary material that defines the various features and gives more details in the selection was added and mentioned in the manuscript.

L 152: “The choice of the features and their definitions are detailed in Supporting Information S4.” 

We believe that including the full definition in the manuscript would impede the readability of the paper. For the control group, the features of both hips were averaged, thus the “pathological” and “contralateral” features are equal. 

Statistics

How were the biomechanical features reduced?

- The description of the PCA was modified as follow to give more details:

L 157: “A principal component analysis (PCA) was performed to keep only the most relevant biomechanical features in the analysis (excluding the time of the phases). The feature were centred and reduced by subtracting the mean and dividing by the standard deviation, to account for the difference of scales. All three groups (patients at M0, patients at M6 and control group) were included in the analysis.”

You state that you kept one feature per component and per subtask. What do you mean by component? PC component? 

- Yes, we mean PC component, it was added to the manuscript for clarity (L 162).

Why do you keep Peak angular velocity thorax and peak vertical velocity thorax in turn to sit which are both associated with PC1 (table 3)?

- Indeed, both features are associated with PC1. The turn-to-sit phase is composed of two movements, turning and sitting, thus although we grouped those two movements in one phase, we wanted to keep a feature per movement. However, we did this for the speed features but not quality feature. This is questionable and thus we removed the feature “peak vertical velocity of thorax” from the analysis for methodological consistency.

Bonferroni correction: what is the number of tests?

- For the differences in features between groups (Table 4), there were 16 tests per group comparison (one for each feature) and 3 group comparisons. The number of tests was 48 leading to a corrected p-value of 0.05/48 = 0.001.

For the comparison of the differences between groups at phase level, there were 4 tests per group comparison and 3 group comparisons. The number of tests was 12 leading to a corrected p-value of 0.05/12 = 0.004.

Reviewer 2 suggested that we use the Holm method instead of the Bonferroni method. We removed the mentions to Bonferroni and used the Holm method instead (L 166).

It is unclear why calculate the percentage of variation. From my understanding of the results, this is not presented anywhere and not related to the aims of the study. Please clarify.

- What we meant by “percentage of variation” was the change in patient’s function between M0 and M6 expressed in percentage of M0. These results are presented in the results, in the section “Evolution of function six months after surgery”. The manuscript was modified as follow to avoid misunderstandings:

L 172: “The change between M0 and M6 was expressed in percentage of the feature value at M0 to express the increase or decrease in function after surgery.”

Results

You often write about a deficit. But it is not explained in the methods how this deficit was defined.

- An explanation of the deficits was added to the method in the Statistics section:

L 169: “A lower function of the patients with respect to the control group was described as a deficit. Depending on the feature, a lower function can be a higher value (e.g. duration of phase) or a lower value (e.g. peak vertical velocity of the thorax for the S2S).”

Numbers in tables, figures and text are repeatedly different from each other even though they should be same (i.e. Fig.2 and Table 4 values for control group).

- The consistency of the numbers in the text, figures and tables were checked.

Why do you use different thresholds for presenting p-values (i.e. Table 4 and Fig 2. once p<0.001, once p<0.003)?

- This is a typo, the correct value was p < 0.001, since the Holm method is now used, it was corrected in Fig. 2 as p < Holm correction. 

You write about a statistically larger deficit in some phases. It is difficult to understand how you derive these results. It would be helpful to directly show the results of the statistical comparison between subtasks (incl p-values and corresponding threshold of Bonferroni correction)

- As suggested by reviewer 2, we are now using the Holm correction, thus we removed mentions to the Bonferroni correction. To clarify, we mentioned in the manuscript, that the deficits were significant with the Holm method (L 214, L 230 and L 236). The Holm method as a varying level of p-value to assess significance of the test and it seemed that reporting both number would impede readability of the manuscript. However, if the reviewer sees fit, we can report both values for the tests of the differences between phases.

Figure 2: The figure caption states that Fig 2 presents an overview about the features selected by the PCA. However, it also contains the duration of the different TUG phases. From my understanding, these were not included in the PCA. Please clarify. 

- Indeed, the duration of the phases was not included in the PCA, we changed slightly the title of Fig2 as follow: “Overview of the duration of the phases and features selected by the PCA for the three groups […]”

Moreover, an explanation on how the points in the plot were selected is missing (sometimes larger number is inside, sometimes outside. How was position of point for M6 defined?).

To avoid having lines crossing each other and make the figure easier to read, we chose to have the control group values on the outside and M0 value on the inside. In this way, an improvement of patients is visually seen as their score by going toward the outside part of the plot. In some cases a lower value will be an improvement (e.g. shorter duration of phase), in other cases a higher value will be an improvement (e.g. higher peak linear velocity of pelvis).

A short explanation was added to the legend of Fig.2: 

Lxx: “Feature values representing higher function are on the outer edge and lower function values are on the inner edge.”

Minor comments:

Abstract

Line 38: Do you mean “improve”? What parameter? Shorter duration of the phases?

- Yes, we changed to ‘improved’. This statement is related to the results with “one global figure per phase” (Table 5). We kept this statement general for the abstract due to the word limitations but if the reviewer thinks it is needed, we can add some more details.

Line 39-41: You conclude that the iTUG could be instrumented with a wearable sensor in a clinical setting. However, from your paper, it’s not clear how that would be achieved. Please revise.

- Three inertial measurement units taped to the thorax, pelvis and thigh on the side of surgery could be used to compute the same features that we computed in this paper. This was mentioned in the discussion in the section “Implementation in clinical settings” (L 327). 

The mention of the Inertial Measurement Units was removed from the abstract to avoid confusion, but kept in the discussion. The abstract was modified as follow:

L 39: “This simple test could be relevant to quantify patient-specific deficits of function to guide and follow post-operative rehabilitation in clinical settings.”

Introduction

Line 47/48: This should be “6 months and 1 year after THA”

- The manuscript was changed accordingly.

Lines 61-62: Please write “test” after each test (6minute walk test, 30s chair stand test or TUG test), or only after TUG test.

- The manuscript was changed accordingly.

Lines 78ff: Please specify what parameters are measured/analyzed in the iTUG.

- The parameters measured/analysed in the iTUG depend on the studies. The point was to say that the iTUG provides biomechanical parameters that can lead to a deeper understanding of the pathologies that are studied. The relevant parameters will differ from pathology to pathology. 

Methods

Lines 107-109: You state that all patients had either stage II, III or IV hip OA, but in Table 1 K/L classification is missing for 4 patients. Why are these data missing?

- Those four patients performed X-rays outside of the Hospital and didn’t bring the X-rays during their visits. It was not possible to evaluate their OA stage with the K/L classification. If the reviewer sees it appropriate, we can mention it in the manuscript.

Line 118: Please change to “destabilised” (you used British English spelling in the rest of the manuscript).

- The manuscript was changed accordingly.

Line 119-120: Please change to “between groups” (or “between populations”)

- The manuscript was changed accordingly.

Line 131: There is a space missing between “chair” and “without”

- The manuscript was changed accordingly.

Lines 133ff: Why did you perform 10 TUGs but only analyze the fastest trial of the first 3 TUGs? Please explain.

- The standard way of performing the TUG test is to perform two tests and keep the fastest time (Podsialdo et al. 1991). Ten measurements were performed for a secondary study that would evaluate the variability of the test. We kept only the first 3 for this analysis to be close to the initial definition of the TUG test and, because for some patients, the measurement of the first or second trial failed. If the reviewer sees it appropriate, we can mention it in the manuscript.

- Podsiadlo D, Richardson S. The Timed “Up & Go”: A Test of Basic Functional Mobility for Frail Elderly Persons. J Am Geriatr Soc. 1991;39: 142–148.

Line 147/S1: Is there any information on what S1 shows? The figure is not self-explanatory, it’s not clear what the sticks indicate.

- We added comments on the first figure of S1 and S2 to guide the reader. If the reviewer sees fit we can add further explanations.

Line 175: Please change to “one global figure per phase”.

- The manuscript was changed accordingly.

Line 176: I think you mean “development” instead of “evolution”. Please change throughout the manuscript (evolution and evolve).

- We modified “evolution” by “change” throughout the manuscript. 

Results (there are no line numbers after page 14, so comments were made on paragraphs)

- Line numbers were added to the whole document.

Table 4: Please check that you use the same precision for mean and SD.

- This was corrected in the table.

Was the SD of mean angular velocity pelvis (turn) and peak angular velocity thorax (turn to sit) really below 1?

- No, this is a typo.

Line 206: Table 6 appears before Table 5 in the text. Please change

- The manuscript was changed accordingly, the number of Table 6 was changed to 5 and conversely.

Line 209: This should be “Bonferroni correction”

- The method to assess the significant differences was changed, the “Holm method” is now mentioned.

Table 5: What do bold numbers indicate? What is parameter corr. total time?

- The bold numbers were used to outline results during our analysis and were removed from the manuscript. 

The parameter “corr. total time” is the coefficient of correlation of the considered feature with the total time of the TUG. It was changed to “R total time” in the table and a description was added to the legend.

Table 6: Please specify in legend, what parameter is meant by percentage.

- The definition of the percentages was done in the Statistics section of the method:

L 168: “To evaluate the average difference of the patients’ function with the control group, the values of the features at M0 and M6 were reported in percentage of the mean of the control group. A lower function of the patients with respect to the control group was described as a deficit. Depending on the feature, a lower function was either a higher value (e.g. duration of phase) or a lower value (e.g. peak vertical velocity of the thorax for the S2S). The change between M0 and M6 was expressed in percentage of the feature value at M0 to express the increase or decrease in function after surgery. Furthermore, the percentage of patients at the level of the control group (i.e. within 1 standard deviation of the control group) was evaluated, as well as the percentage of patients with an increase in function between M0 and M6. The percentages of differences with the control group and the percentage of change between M0 and M6 were correlated with the differences and change of the total time.”

The legend of the table was modified as follow for clarity:

“Table 5: Mean difference ±SD (95% confidence interval) between groups at phase level expressed in percentages”

Table 5/6: Were the 95% CI of the difference really that much smaller than the SD of the difference?

- Yes, we used the formula 95%CI = mean ± 1.96 * SD / sqrt(n) with n the number of patients. As we had 71 patients it reduces the expected range of the 95% CI.

As an example, for the values of the S2S comparison of patients M0 vs CG, the mean and SD of the difference for the 71 patients are respectively -30% and 37% which lead to:

- A lower bound of: -30 – 1.96 * 37 / sqrt(71) = -39

- An upper bound of: -30 + 1.96 * 37 / sqrt(71) = -21

P.14, paragraph 1, last sentence (49-86%): why do you here include all parameters and in previous paragraph only those with significant differences?

- This is a mistake, we modified the manuscript as follow:

L 230: “Only 27% of patients had a range of hip flexion-extension during walking at the level of the controls while the other significant features had 46% to 62% of patients at this level (Table 5).”

Discussion

P.15, paragraph 2: what scores of the patients? This implies to me that you compared PC scores, but from the results it’s not what you did. Please clarify.

- With “scores of the patients”, we meant the values of the features for the patients. As this might be misleading, we modified the manuscript as follow:

L 259: “To that end, the TUG test was divided into four phases [15] representing different daily living activities (sit-to-stand, walking, turning and turn-to-sit), multiple features were selected by PCA and finally, the features of the patients were compared before surgery and at 6 months after surgery and to a control group for each phase.”

P.15, paragraph 3, last sentence: You state that this (hip flex/ext range walking phase) has a strong positive effect. However, only 20% of patients reach level of controls. Isn’t this contradictory?

- We meant that this was the feature with the largest number of patients with an increase (82%). We modified this part of the discussion as follow:

L 269: “This feature had the lowest number of patients at the level of the control groups before and 6 months after surgery (8% and 27% respectively). However, this feature showed the highest number of patients with a positive change in function 6 months after surgery (82%). This suggests that, despite the deficits 6 months after surgery, THA has a positive effect on the sagittal hip kinematics during walking.”

P.17, variability of outcomes: From the presented results, I don’t understand how you analysed variability.

- The SD values on the percentage of deficits at feature level and phase level were very high. The SD values were often at the same level as the mean and even larger than the mean for some cases (Table5 and 6). In our opinion, this indicates a high variability in the profiles of the patients.

This idea was clarified in the manuscript:

L 299: “However, the SD levels of the percentage of differences between groups where very high, e.g. the SD of the deficits of the total TUG time at M0 was 38% while the mean deficits was 39%. These high levels of SD suggests the high variability of profiles in patients. Indeed, some patients were at the level of control groups while others did not improve their function at M6 (Supporting Information S6).”

P.17, paragraph 1: were there any patients that had decreased performance 6 months postoperative?

- Yes, some patients had decreased performance in multiple feature. We did not see patients with a clear decrease on all feature. The supplementary file S6 shows the values of the features at M0 and M6 of all patients compared to the control groups. As examples, patient 22 had a decrease in the duration of most phases and patient 42 had a decrease on most of the features. On the contrary, patient 7 had an increase for most of the features.

P.17, paragraph 2: please change “strong post-surgery deficit” to “large post-surgery deficit”

- The manuscript was changed accordingly.

P.18, paragraph 1: Please change to “three synchronized IMUs…”

- The manuscript was changed accordingly.

P.18, paragraph 2: you state that a simple 1 IMU/smartphone provides enough parameters to assess the function of the patients. However, from your analysis this is not valid. The presented parameters were not measured with IMUs and would need several sensors (data on pelvis, hip, thorax…). Please elaborate on how such a sensor should be set up according to your results.

- To obtain the same features we would need IMUs on thorax, pelvis and thigh of the pathological side. However, this setup could be cumbersome in clinical settings and we suggest that a setup with one IMU would have more chances to be used in such settings. We expect that one IMU on the thorax could provide sufficient information to assess the function of the patients, but it could not provide the same features as the one described in this study. We removed this part from the manuscript as it is more a speculation and because the present paper does not have results supporting this claim.

P.18, paragraph 2, last 4 lines: Please use either “a test” or “tests”. Please change to “However, further studies are needed…”

- The manuscript was changed accordingly.

P.18, paragraph 4. Please change to “duration of the phases” instead of “time of the phases”.

- The manuscript was changed accordingly.

P.19, paragraph 1: You state that there was a study on minimal detectable changes of iTUG parameters. How do the difference between pre and post patients compare to these minimal detectable changes? Please elaborate.

- The sentence we used could lead to misinterpretation. The study by Smith et al. (2020) reported in the paper assessed only the MDC of the total time evaluated measured with a commercial iTUG system but did not report MDC for the parameters of the TUG. However, they reported the effect size “to quantify the change observed in each of the [i]TUG parameters”. We modified the sentence in the manuscript as follow:

L 336: “Recently, the responsiveness to change of multiple parameters of the iTUG and minimal detectable change of the total time have been evaluated among older adults (63.9 ± 6.1 years) undergoing a six-week physiotherapy program with encouraging results [36]. Indeed, the responsiveness to change was shown with small to moderate effect sizes and the MDC of the total time was 0.77s.”

P.20, paragraph 1: You abbreviated timed-up and go test as TUG test, please change.

- The manuscript was changed accordingly. 

Reviewer #2

The study explores functional deficits before and its change after total hip arthroplasty with an instrumented timed up-and-go test, analyzing phase specific differences between patients and a control group. The topic is of clinical relevance and the sample size adequate to produce meaningful results. The relevance and experimental setup are well described and introduced. Limitations are within the study design, where no post-measurement of the control group is available, resulting in multiple testing and a reduction in statistical power. Further methodological shortcomings, including the variable selection, reduce the significance of the results. In some cases, false or non-ideal statistical tests are used and in other cases (e.g. PCA), methods are not well enough described to judge the correctness of the results. Combined with further language and typing errors, major revisions are necessary to improve the manuscript. Nevertheless, the study has the potential to come to good and meaningful results.

Major Revisions: To improve the study/manuscript following main issues should be revised:

1. The preselection of variables is arbitrary and not theory or hypothesis driven. The sole explanation are previous studies, who investigated other disorders and included further variables which have not been considered in the current study (e.g. spatiotemporal variables, such as cadence, stride length, length of single support…). Ideally the variables should be based on known limitations in hip OA patients, identified in previous studies. Furthermore, the joint angles over the complete range of motion (ROM) are unspecific, making conclusions about deficits for specific movements or walking phases difficult. For example, dividing hip ROM in flexion and extension would help to interpret whether deficits during walking are more related to the late stance phase (restricted hip extension), indicating problems to generating step length or swing phase (restricted hip flexion), which might result in higher fall risk due to lower foot clearance. It is unclear why peak thorax obliquity was only analysed during sit2stand/turn2sit and not walking/turning and why peak thorax obliquity was not directly investigated of the pathological side? Conversely, what is the rationale behind investigating different mean velocities of the pelvis and thorax. In which scenario do the authors expect to find notable differences in the mean velocities of these segments?

- A table was added to supporting information with more details on the choice of the features.

The spatiotemporal parameters of gait were not selected due to the very short distance of walking that was below or equal to 3m. The target for turn was at 3m from the table, but, when considering that some participants started turning before the target and some started walking while standing up, the distance of walking was often below 3m. Previous studies showed that 5 steps are required to have stable variability of gait parameters [1] and a modified TUG with 7m of walking was recommended to assess spatiotemporal parameters during iTUG [2]. However a distance of 2.4m to 3m was found sufficient to measure gait speed [3]. 

Regarding the hip ROM of flexion-extension, this parameters was selected as it is commonly used in THA literature [4]. We agree that dividing the ROM in flexion and extension would be an interesting to understand patient’s limitation but that it would be more appropriate for a specific study focused on gait.

Regarding thorax and pelvis velocities, iTUG studies sometimes use angular/linear velocities of the pelvis [5] (or waist [6]), and sometime velocities of the thorax [2]. We chose to take both in the analysis and to identify the most relevant in this specific study with PCA.

1. Sangeux M, Passmore E, Graham HK, Tirosh O. The gait standard deviation, a single measure of kinematic variability. Gait Posture. 2016;46: 194–200. doi:10.1016/j.gaitpost.2016.03.015

2. Salarian A, Horak FB, Zampieri C, Carlson-kuhta P, Nutt JG, Aminian K. iTUG , a Sensitive and Reliable Measure of Mobility. IEEE Trans neural Syst Rehabil Eng. 2010;18: 303–310. 

3. Stuck AK, Bachmann M, Füllemann P, Josephson KR, Stuck AE. Effect of testing procedures on gait speed measurement: A systematic review. PLoS One. 2020;15. doi:10.1371/journal.pone.0234200

4. Bahl JS, Nelson MJ, Taylor M, Solomon LB, Arnold JB, Thewlis D. Biomechanical changes and recovery of gait function after total hip arthroplasty for osteoarthritis : a systematic review and meta-analysis. Osteoarthr Cartil. 2018;26: 847–863. doi:10.1016/j.joca.2018.02.897

5. Mellone S, Tacconi C, Chiari L. Validity of a Smartphone-based instrumented Timed Up and Go. Gait Posture. 2012;36: 163–165. doi:10.1016/j.gaitpost.2012.02.006

6. Zakaria NA, Kuwae Y, Tamura T, Minato K, Kanaya S. Quantitative analysis of fall risk using TUG test. Comput Methods Biomech Biomed Engin. 2015;18: 426–437. doi:10.1080/10255842.2013.805211

2. PCA:

The PCA might be a good approach to identify important variables. However, insufficient clarity and scarce description make it impossible to judge the results. Especially in the presence of differently scaled variables the standardization and re-scaling is of utmost importance for reliable results. It is unclear how the variables were centered and what is meant by “reduced”. Please clarify.

It is not described if both groups were included in the PCA analysis.

-The description of the PCA was modified as follow to give more details:

L 157: “A principal component analysis (PCA) was performed to keep only the most relevant biomechanical features in the analysis (excluding the time of the phases). The feature were centred and reduced by subtracting the mean and dividing by the standard deviation, to account for the difference of scales. All three groups (patients at M0, patients at M6 and control group) were included in the analysis.”

Why was the time not included? Wouldn’t that give important information which quality features are related to the time? Doing so might also spare the correlations, which further complicate the analysis.

The duration of the phases are the most commonly reported parameters in iTUG studies. We chose beforehand to keep them in the analysis to be able to compare our results to the literature. Thus, the PCA guided only the selection of biomechanical features.

The choice of variables based on the cos² values is inconsistent. For example, in quality during walking “Range of flexion contralateral hip (0.47)” and “Range obliquity thorax (0.30)” have higher cos² value than “RMS obliquity thorax (0.27)”. The selection method or the description needs clarification.

For the quality during walking, the selected feature was “Range obliquity thorax (0.30)” and not “RMS obliquity thorax (0.27)”. The bold font is a typo in the Supplementary file. The “range of flexion contralateral hip” does have a higher cos2 than the range of obliquity of the thorax but it is on the same PC as the “range of flexion of pathological hip” which is why it was excluded. In the first version of the manuscript we selected one feature per component apart for the turn-to-sit phase that regroups two movements. However, as suggested by reviewer 1, we removed the peak vertical velocity of the thorax for consistency in the method.

After checking the PCA results the feature “Distance chair to start of turn” of the turn to sit phase was added to the analysis to have a consistent selection of parameters.

3. The Kolmogorov-Smirnoff test is inappropriate to compare the patient characteristics. Age distribution can be the same in a group of children and adults. The interesting information is, whether the patient and control group are of similar age. Therefore, t-tests are needed, and the K-S tests only serves to test the assumption of normal distribution for the t-tests. For weight, a non-parametric test like the Mann-Whitney U test is necessary.

- Unpaired Student’s t-tests were added to table 1 to assess the differences between groups. However, we do not understand why the test for the weight should be non-parametric. Could the reviewer develop on this point?

4. Number of tests and p-value correction:

The fact that the study design does not allow for repeated measures of variance and the downside of multiple testing is a problem. The current approach with the Bonferroni method is very conservative: if I count right 16*3 t-test were calculated for the variables plus 4 for the phases, resulting in a significance level of 0.05/52 <= 0.001, which conversely inflates the false negative (type II error) risk.

I would recommend controlling the FDR with the Benjamini-Hochberg correction or the Holm method.

- The Holm method was used instead of the Bonferroni correction and all data updated. As expected, this correction included more significant differences (see Table 4):

- Difference in range of thorax obliquity during walking between M0 and M6

- Difference in duration of turn between M0 and M6

- Difference in range of thorax obliquity during turn to sit between M0 and CG

 At phase level, we obtained the following modifications:

- Walking had a significantly larger deficit than the three other phases at M0

- Walking had a significantly larger deficit than the S2S and turn phases at M6

These new results strengthen the conclusion of the first version of our analysis, i.e. walking seems to present the largest deficit before and 6 months after surgery for patients undergoing THA.

To further strengthen the study effect sizes with CI could be calculated.

Effect sizes and 95% Confidence Interval were added to Table 4.

5. Overall deficit in percent:

The approach to calculate the overall deficit only on the significantly different variables within a phase is questionable. What if there was no significant variable within a phase? Has this phase gets no overall rating? What about cases where significant differences exist only pre- or post-surgery? Is the variable only included in one of the overall scores but not in the other?

Overall score should include all important variables (already identified by the PCA) and differences tested afterwards.

- From our point of view, if the feature was not significantly different, there was no deficit/difference to investigate which is why we only took into account the significant features. If there was no significant variable, then there would be no percentage of deficit/difference. However, this method will increase the average deficit of the considered phase. 

The point of view of the reviewer seems to be more robust from a methodological standpoint. We implemented these recommendations in the paper and modified the Statistics section of the method as follow:

L 179: “Finally, the percentages of the features selected with PCA were averaged for each phase to have one global figure per phase.”

Further minor revisions:

Note: These corrections are not exhaustive since it can be expected that major changes to the manuscript will follow the main revisions. Therefore, the corrections are rather examples hinting towards deficits requiring attention.

1. Further remarks:

• “52 aged-matched controls” if they were age-matched, there would not be a 3y difference in age.

- The mention “aged-matched” was removed from the manuscript.

• “pregnancy or breast-feeding patients” relevant in this age group?

- It is not relevant for the patients of this study. However, during the recruitment of patients, the range of age was between 25 and 85 years old which made this criterion relevant. For this new version of the manuscript we mentioned the age at the start of the exclusion criteria:

L 110: “Exclusion criteria included patients younger than 25 years old or older than 85 years old”

If the reviewer prefers, we can remove the mention of “pregnancy or breast-feeding patients” from the manuscript.

• “The patients and control groups did not differ in sex, […]” as pointed out earlier, the K-S test does only test distribution differences and gives no insight about mean group differences.

- An unpaired Student’s t-test was added to the analysis and did not show significant differences. 

• Table 4: Please revise/double-check. SD of Mean angular velocity pelvis (deg/s) (turn) and Peak angular velocity thorax (deg/s) (turn2sit) at M6 appear wrong.

- The consistency of the numbers in the text, figures and tables were checked an updated when needed.

• Use the same number of digits for a variable mean and SD.

- The manuscript was changed accordingly.

• Table 5 is inconsistent. Sometime coloring is false (e.g. SD from Range thorax obliquity (%) during walking pre-post comparison). Also reporting of correlations is inconsistent (sometimes left out despite sig. diff and sometimes displayed without sig. diff.)

- The consistency was checked and results were modified accordingly.

• “This deficit was statistically larger than the other phases (p < 0.05) but not with the Bonferonni correction.“ Reporting „significant“ results, which are insignificant after p-value correction goes against the logic of doing a correction.

- The manuscripts now refers only to the Holm method.

• “[…] selected by PCA and the scores of the patients […]” implies that PCA scores were compared, which is not the case.

- This was not the case indeed. The “score” here relates to the features and not to the values of the PCA. This seems to be a formulation issue, we modified the sentence:

L 260: “multiple features were selected by PCA and finally, the features of the patients were compared …”

• “[…] presence of a limp (indicated by the range of thorax obliquity).” Limping might not be very well represented by the thorax ROM. It does not reveal if the trunk lean was one-sided. Furthermore, thorax ROM might not be a good measure of limping because there are probably ways of limping that don’t involve an excessive trunk lean. Letter is a good strategy to unload the hip and is a known compensatory movement for weak hip abductors or hip pain. For limping, however, a better and more direct measure would be the single support phase.

- We changed “limp” to “compensatory movements” to be more general.

• The section “Implementation in clinical settings” might be well-meant, however, has too little relation to the results.

- This study was a first step toward the development of clinical tools for the measurements of function. Thus, we believe that this section can be of interest to provide a broader context and expected applications of the study to the reader. However, if the reviewer thinks this is section should be left out we can remove it from the paper.

• “The reliability, validity and responsiveness should be assessed for the time of the tasks and for the features of this study before using them in clinical practice, […]” Limitations should relate to the study.

- This part was moved to the section “Implementation in clinical settings”.

• Last paragraph of Limitations: “Surprisingly, the PCA identified parameters…” is not a limitation and belongs in the discussion.

- This paragraph was moved to the section “Variability of outcomes” where the PCA is discussed.

2. Variable names:

The variables should be named more precisely.

“Lateral distance between feet”. Is that step width? There is no lateral distance between the feet.

- This parameter is the distance between the midpoint of the 2 markers of the right and left foot in the lateral direction of the lab reference frame when the patient is sitting in the chair. It was changed to “Width base of support”. A thorough description of the features was added in Supporting Information S4.

 “Only 8% of patients had a range of hip flexion extension during walking […]” variable name in tables implies only hip flexion was analyzed. If the variable comprises hip flexion/extension it should be named accordingly.

- This variable takes into account hip flexion and extension. It was modified throughout the manuscript. 

3. Language and typos

Language and wording are not quite yet on a publication level and should get revised by a native speaker or expert.

Examples for awkward phrasing, wording, grammar, and typos:

• “Exclusion criteria included a previous surgery to the hip” � previous hip surgery.

- The manuscript was changed accordingly.

• “hospital length of stay after surgery” � hospitalization?

- This term is commonly used in medical journals, it represents the time from the day of admission to the day of discharge. See reference [12] in the manuscript: Petis et al. (2016).

- Petis SM, Howard JL, Lanting BA, Somerville LE, Vasarhelyi EM. Perioperative Predictors of Length of Stay After Total Hip Arthroplasty. J Arthroplasty. 2016;31: 1427–1430. doi:10.1016/j.arth.2016.01.005

• “interfere or be destabilised by gait analysis” � interfere with or be worsened by …

- The manuscript was changed accordingly.

• “(i.e. above the mean of the control group minus one standard deviation)” � within 1SD of the control group

- The manuscript was changed accordingly.

• “Evolution” is the wrong word to describe the change/difference between pre/post

- We modified “evolution” by “change” throughout the manuscript.

• “pace” unusual in this context. Speed is probably more appropriate.

- We changed “pace” by “speed” as suggested.

• “TUG test was not performed previously for patients with end stage” � previously in patients.

- The manuscript was changed accordingly.

• “up-an-go” � up-and-go.

- The manuscript was changed accordingly.

In conclusion, there is much room for improvement and besides the major issues all the little errors throughout the manuscript and tables indicate that the manuscript is not ready for publication. Nevertheless, the study might provide very useful information to the scientific community.

---

## [Decision Letter · Decision Letter 1]

5 May 2021

PONE-D-20-38214R1

Which functional tasks present the largest deficits for patients with total hip arthroplasty before and 6 months after surgery? A study of the Timed Up-and-Go phases

PLOS ONE

Dear Dr. Gasparutto,

Thank you for submitting your manuscript to PLOS ONE. After careful consideration, we feel that it has merit but does not fully meet PLOS ONE’s publication criteria as it currently stands. Therefore, we invite you to submit a revised version of the manuscript that addresses the points raised during the review process.

We look forward to receiving your revised manuscript.

Kind regards,

Peter Andreas Federolf

Academic Editor

PLOS ONE

Journal Requirements:

Reviewers' comments:

Reviewer's Responses to Questions

**Comments to the Author**

1. If the authors have adequately addressed your comments raised in a previous round of review and you feel that this manuscript is now acceptable for publication, you may indicate that here to bypass the “Comments to the Author” section, enter your conflict of interest statement in the “Confidential to Editor” section, and submit your "Accept" recommendation.

Reviewer #1: (No Response)

Reviewer #2: All comments have been addressed

2. Is the manuscript technically sound, and do the data support the conclusions?

Reviewer #1: Yes

Reviewer #2: Yes

3. Has the statistical analysis been performed appropriately and rigorously? 

Reviewer #1: Yes

Reviewer #2: Yes

4. Have the authors made all data underlying the findings in their manuscript fully available?

Reviewer #1: Yes

Reviewer #2: Yes

5. Is the manuscript presented in an intelligible fashion and written in standard English?

Reviewer #1: No

Reviewer #2: No

6. Review Comments to the Author

Reviewer #1: The manuscript was greatly improved and the performed analysis is much clearer now. Even though there is so much information and data presented in the manuscript that it’s not always easy to follow the presented data. The description of the statistical methods still needs some minor improvements to make it clearer to the reader what tests were performed. There are also still some grammatical errors (use of singular/plural) and the manuscript should be checked in that regard again.

Abstract:

The performed test should be named “timed up-and-go test” in the abstract and manuscript text and not only “timed up-and-go”. (lines 24, 30 and 136)

Line 26: It should say: “biomechanical analysis of its phases was previously used…”

Introduction:

Line 68: This should be changed to “people diagnosed with hip OA”. Hip arthroplasty is not a diagnosis.

Methods

Using a t test to test for differences in sex distribution is not very common. I think Chi-square test would be more appropriate to check whether the patient and control group had different distribution of sex.

Line 178: Which correlation coefficient was calculated? Pearson? It appears that only significant correlations were reported in the results. What was the level of significance for the correlation coefficients?

Line 179: Do you mean “the percentage differences of the features selected by PCA”?

Line 180-182: It’s not entirely clear what you tested here. The sentence implies that you used paired t tests to test for differences between patients and controls which was not the case. This should be clarified. Maybe something like this would be clearer (if this is indeed the intended meaning): “Paired Student’s t tests were performed to assess whether the average differences with respect to the control group or between M0 and M6 differed significantly between the different phases of the TUG .”

Results:

Line 232: It is unclear what is meant with “other significant features”. Do you mean “features with significant differences”?

Line 235: Please change to “with the Holm method.”

Line 248: correlation of deficit in total and peak walking speed not presented in table 6 (it would be logical but peak walking speed is never presented throughout the paper).

Line 249: “the step numbers” should be changed to “number of steps”.

Discussion

Line 284: “time of the walking phase” should be changed to “duration of the walking phase”

Lines 281-290: It’s confusing that you’re talking about correlations between total time and features but actually mean the correlations between the change in total time and change in features (which is what the title implies and what is presented in the results). For a better understanding for the readers it should be specified in the text as well.

Lines 321-324: It is unclear what the authors want to say with that. PCA selects features with the highest variations but that does not mean that there need to be significant differences between groups. So, it’s no surprise to me that some of the selected features are not significantly different between groups. The second point about the measurement accuracy is important but should be discussed in a different context (i.e. are the statistically significant differences also clinically meaningful).

Line 363: there is a comma too much “…and change, but a categorical feature…”

General points on grammar and use of abbreviations. The list might not be complete so it should be checked again

Singular/plural of nouns are not always used correctly:

o Line 158: the features were…

o Line 301: the mean deficit was…

Verbs are not always in the correct form (3rd person singular -> is/was/suggests…; 3rd person plural -> are/were/suggest)

o Line 252: should be “the change… was associated…”

o Line 264: “the walking task represents the main limitation”

o Line 285/286: “total time could be good indicator… but does not seem…”

o Line 301: “these high levels of SD suggest…”

o Line 316: “the reduction of pain… leads…”

Introduced abbreviations are not consistently used or later abbreviated or never introduced

o OA: line 86, line 332, line 349

o TUG: line 367, line 372

o THA: line 73, line 296, line 366

o SD: abbreviation should be introduced at first use (line 108), but then used in text (i.e. lines 159, 175)

o S2S (line 172) is never explained in text (only in figures) and otherwise not used in abbreviated form in the text. For better readability I’d suggest writing it out here (sit to stand?) and not use the abbreviation

o MDC in line 340 is not introduced

Reviewer #2: Thank you for implementing the changes. Following are responses and further remarks.

Authors:

“The spatiotemporal parameters of gait were not selected due to the very short distance of walking that was below or equal to 3m. The target for turn was at 3m from the table, but, when considering that some participants started turning before the target and some started walking while standing up, the distance of walking was often below 3m. Previous studies showed that 5 steps are required to have stable variability of gait parameters [1] and a modified TUG with 7m of walking was recommended to assess spatiotemporal parameters during iTUG [2]. However a distance of 2.4m to 3m was found sufficient to measure gait speed [3]. Regarding the hip ROM of flexion-extension, this parameters was selected as it is commonly used in THA literature [4]. We agree that dividing the ROM in flexion and extension would be an interesting to understand patient’s limitation but that it would be more appropriate for a specific study focused on gait. Regarding thorax and pelvis velocities, iTUG studies sometimes use angular/linear velocities of the pelvis [5] (or waist [6]), and sometime velocities of the thorax [2]. We chose to take both in the analysis and to identify the most relevant in this specific study with PCA.”

Well, many TUG studies investigate the spatiotemporal parameters and, in my opinion, it makes sense. For example, the number of steps, step time, single support length, step length etc. can be very informative in terms of patient agility and potential limping.

How where the kinematics calculated for the different phases? Was it the mean of all steps within the phase for each side or both sides? Furthermore, only the fastest trial was chosen. Why not the mean of the 10 trials? Especially, as you mention the study of Sangeux [1], who did not really investigate the variance of consecutive steps but rather number of trials. They recommend 6 trials for kinematics. Based thereon, the results of the current study would be questionable as well, given that only one TUG trial was analyzed. Given that the gait kinematics most likely deviate from normal around the phase transitions.

[2] did not report differences in spatiotemporal parameters between the TUG (3m) and iTUG (7m), despite the time. A recommendation was not given. Only the conclusion that spatial parameters are less reliable due to the method (accelerometer/gyroscope).

[4] reported ROM probably out of convenience from max. flexion and extension. The included studies, such as, Klausmeier V. (2010) and Mayr E. (2009) report the flexion/extension separately, which makes a lot of sense.

[5] and [6] are accelerometer/smartphone studies and the velocity site is most likely defined by the limited application possibilities.

In conclusion, the variable choice remains inconclusive, and a good rationale is missing. On the one hand, the authors identify the most relevant variables using the PCA but on the other hand, cherry pick and limit their pre-selection to variables which clearly stem and are defined by the convention/limitations of accelerometer or IMU systems.

Two approaches would be conclusive:

1. Use ALL variables previously used in TUG tests to be able to compare results to accelerometer based iTUGs and to identify the most relevant.

2. Use evidence-based variables, which were previously identified to be relevant for THA patients.

Authors: “- Unpaired Student’s t-tests were added to table 1 to assess the differences between groups. However, we do not understand why the test for the weight should be nonparametric. Could the reviewer develop on this point?”

The K-S test is used to test the assumption of normal distribution a t-test. A significant result in the K-S test indicates a non-normal distribution and a violation of the t-test assumption. In this case a non-parametric test is warranted.

Commonly the scientific community assumes that the researchers know and use the appropriate assumption tests for their statistical analysis and usually do not ask to report the assumption test results. The K-S test results can be omitted in the table, but it should be indicated that a non-parametric test is used to compare the weight.

Authors: “If the reviewer prefers, we can remove the mention of “pregnancy or breast-feeding patients” from the manuscript.”

If the authors think it is relevant, it can remain in the manuscript.

Reviewer:

• The section “Implementation in clinical settings” might be well-meant, however, has too little relation to the results.

Authors:

- This study was a first step toward the development of clinical tools for the measurements of function. Thus, we believe that this section can be of interest to provide a broader context and expected applications of the study to the reader. However, if the reviewer thinks this is section should be left out we can remove it from the paper.

Reviewer: Since it was not subject of the current investigation, I would exclude this part.

Reviewer:

• “hospital length of stay after surgery” hospitalization?

Authors:

- This term is commonly used in medical journals, it represents the time from the day of admission to the day of discharge. See reference [12] in the manuscript: Petis et al. (2016).

-Petis SM, Howard JL, Lanting BA, Somerville LE, Vasarhelyi EM. Perioperative Predictors of Length of Stay After Total Hip Arthroplasty. J Arthroplasty. 2016;31: 1427–1430. doi:10.1016/j.arth.2016.01.005

Reviewer: It remains a clumsy wording and weakens the readability but since it is minor issue the authors may choose to stick to their wording.

Further minor corrections:

Methods:

“The level of statistical significance was controlled with the Holm method (� = 0.05)”

Better: To account for multiple testing the Holm’s method was used to adjust the p-values.

L119: Student t-tests change to Student’s t-test

P17L235: “although not statistically significant with the Holm.”

Was it significant different without the Holm correction? Either it is significant or not. Either the authors acknowledge the method or not.

P17L235: a similar increase.

P18L269ff: “This feature had the lowest number of patients at the level of the control groups before and 6 months after surgery (8% and 27% respectively).”

Hard to comprehend. Please rephrase.

P19L281: with a strong

P19L285ff.: “The total time could be a good indicator of speed parameters but do not seem to reflect as well the quality of movement such as the presence of compensatory movements (e.g. high range of thorax obliquity).”

Check for errors, maybe rephrase.

P20L305: “help to identify patients at risks of lowering their level of function after surgery” Appears wrong: implies that patients actively lower their function but it’s something that happens to them.

P20L307: “high variability observed with the analysis of the TUG”…the variability is a result of the analysis and not an observation of the analysis. Please revise the sentence/wording.

P22L348ff.: “The patient population and control group had a significant difference in term of weight, but a higher weight was reported as a risk factor for hip osteoarthritis [43] which could explain this difference.”

Is not really a limitation, at least not how it is stated. Weight might be a limitation in terms of different kinematics or measurement inaccuracies due to marker placement or skin artifacts.

P22L360ff:

“The feature “distance chair to start of turn” was chosen with the assumption that a longer distance would imply that the patients turned and sat at the same time, i.e. with an overlapping transition strategy. This feature was continuous, as this study aimed at assessing percentages of differences and change, but, a categorical feature classifying the strategy of the patients could be of interest to understand the patient’s function.”

Why is this a limitation?

7. PLOS authors have the option to publish the peer review history of their article (what does this mean?). If published, this will include your full peer review and any attached files.

Reviewer #1: No

Reviewer #2: No

---

## [Author Response · Author response to Decision Letter 1]

28 Jun 2021

Reviewer #1 

The manuscript was greatly improved and the performed analysis is much clearer now. Even though there is so much information and data presented in the manuscript that it’s not always easy to follow the presented data. The description of the statistical methods still needs some minor improvements to make it clearer to the reader what tests were performed. There are also still some grammatical errors (use of singular/plural) and the manuscript should be checked in that regard again.

- The paper was proofread by a native English speaker: Darren Hart of “publish-or-perish.ch”, a professional proofreader.

Abstract:

The performed test should be named “timed up-and-go test” in the abstract and manuscript text and not only “timed up-and-go”. (lines 24, 30 and 136)

- The manuscript was modified accordingly.

Line 26: It should say: “biomechanical analysis of its phases was previously used…”

- The manuscript was modified accordingly.

Introduction:

Line 68: This should be changed to “people diagnosed with hip OA”. Hip arthroplasty is not a diagnosis.

- The manuscript was modified accordingly.

Methods

Using a t test to test for differences in sex distribution is not very common. I think Chi-square test would be more appropriate to check whether the patient and control group had different distribution of sex.

- The t-test was changed to a Chi-square test, there is still no significant differences.

Line 178: Which correlation coefficient was calculated? Pearson? It appears that only significant correlations were reported in the results. What was the level of significance for the correlation coefficients?

- Yes, we used Pearson correlations and reported only the significant correlations. The Holm method was used to adjust the level of significance. The manuscript was modified as follow:

L177: “Pearson correlations were calculated between the total TUG time and selected features for percentage differences to the control group and the percentage of change between M0 and M6. The level of significance was adjusted using the Holm method (� = 0.05) [39]. Only significant correlations were reported.”

Line 179: Do you mean “the percentage differences of the features selected by PCA”?

- The manuscript was modified for clarity. See previous comment.

Line 180-182: It’s not entirely clear what you tested here. The sentence implies that you used paired t tests to test for differences between patients and controls which was not the case. This should be clarified. Maybe something like this would be clearer (if this is indeed the intended meaning): “Paired Student’s t tests were performed to assess whether the average differences with respect to the control group or between M0 and M6 differed significantly between the different phases of the TUG .”

- Thank you, your sentence is clearer. The manuscript was modified as suggested.

Results:

Line 232: It is unclear what is meant with “other significant features”. Do you mean “features with significant differences”?

- Yes, this is what we meant. The manuscript was modified as suggested.

Line 235: Please change to “with the Holm method.”

- The manuscript was modified accordingly.

Line 248: correlation of deficit in total and peak walking speed not presented in table 6 (it would be logical but peak walking speed is never presented throughout the paper).

- Indeed, in this study, the feature corresponding to the peak walking speed is the peak forward velocity of the pelvis. This was changed in the manuscript.

L247: “The deficits and changes in total TUG time were mainly associated with speed parameters, especially with the deficits and changes in the duration of the walking phase and with the peak forward velocity of the pelvis (Table 6).”

Line 249: “the step numbers” should be changed to “number of steps”.

- The manuscript was modified accordingly.

Discussion

Line 284: “time of the walking phase” should be changed to “duration of the walking phase”

- The manuscript was modified accordingly.

Lines 281-290: It’s confusing that you’re talking about correlations between total time and features but actually mean the correlations between the change in total time and change in features (which is what the title implies and what is presented in the results). For a better understanding for the readers it should be specified in the text as well.

- The title and section were modified as follow:

L 245:“ Associations between deficits and changes in features and deficits and changes in total TUG times (Table 6)

The deficits and changes in total TUG time were mainly associated with speed parameters, especially with the deficits and changes in the duration of the walking phase and with the peak forward velocity of the pelvis (Table 6). The deficits in the range of flexion–extension of the patient’s pathological hip during walking at M0 and M6, the deficit in the number of steps taken during the turning phase at M6, and the deficits in the distance from the chair to the start of the turn at M0 and M6 were the only quality parameters associated with the difference in total time that displayed moderate correlations (Table 6). The change in total TUG time between M0 and M6 was only associated with speed parameters. The highest correlations were with the change in the time of the walking phase and the change in peak walking speed.”

Lines 321-324: It is unclear what the authors want to say with that. PCA selects features with the highest variations but that does not mean that there need to be significant differences between groups. So, it’s no surprise to me that some of the selected features are not significantly different between groups. The second point about the measurement accuracy is important but should be discussed in a different context (i.e. are the statistically significant differences also clinically meaningful).

- The PCA discussion was removed and the point about measurement accuracy was added in the limitation section: 

L 354: “The duration of test phases and the time differences between groups were relatively small, especially for the sit-to-stand and turning phases. Thus, those results should be taken with care as the difference could be within measurement accuracy of the algorithm. Moreover, data centring and reduction led to the selection of parameters with non-clinically meaningful differences. Indeed, the PCA identified parameters with differences that were within the range of measurement accuracy (1 degree), e.g. the peak thorax flexion and range of thorax obliquity during the sit-to-stand phase. Future studies could use measurement accuracy as a criterion during the feature selection.”

Line 363: there is a comma too much “…and change, but a categorical feature…”

- The manuscript was modified accordingly.

General points on grammar and use of abbreviations. The list might not be complete so it should be checked again

Singular/plural of nouns are not always used correctly:

o Line 158: the features were…

- The manuscript was modified accordingly.

o Line 301: the mean deficit was…

- The manuscript was modified accordingly.

Verbs are not always in the correct form (3rd person singular -> is/was/suggests…; 3rd person plural -> are/were/suggest)

o Line 252: should be “the change… was associated…”

- The manuscript was modified accordingly.

o Line 264: “the walking task represents the main limitation”

- The manuscript was modified accordingly.

o Line 285/286: “total time could be good indicator… but does not seem…”

- The manuscript was modified accordingly.

o Line 301: “these high levels of SD suggest…”

- The manuscript was modified accordingly.

o Line 316: “the reduction of pain… leads…”

- The manuscript was modified accordingly.

Introduced abbreviations are not consistently used or later abbreviated or never introduced

o OA: line 86, line 332, line 349

- The manuscript was modified accordingly.

o TUG: line 367, line 372

- The manuscript was modified accordingly.

o THA: line 73, line 296, line 366

- The manuscript was modified accordingly.

o SD: abbreviation should be introduced at first use (line 108), but then used in text (i.e. lines 159, 175)

- The manuscript was modified accordingly.

o S2S (line 172) is never explained in text (only in figures) and otherwise not used in abbreviated form in the text. For better readability I’d suggest writing it out here (sit to stand?) and not use the abbreviation

- The manuscript was modified accordingly.

o MDC in line 340 is not introduced

- MDC stands for minimal detectable change. The abbreviation was removed from the manuscript.

Reviewer #2: 

Thank you for implementing the changes. Following are responses and further remarks.

Authors:

“The spatiotemporal parameters of gait were not selected due to the very short distance of walking that was below or equal to 3m. The target for turn was at 3m from the table, but, when considering that some participants started turning before the target and some started walking while standing up, the distance of walking was often below 3m. Previous studies showed that 5 steps are required to have stable variability of gait parameters [1] and a modified TUG with 7m of walking was recommended to assess spatiotemporal parameters during iTUG [2]. However a distance of 2.4m to 3m was found sufficient to measure gait speed [3]. Regarding the hip ROM of flexion-extension, this parameters was selected as it is commonly used in THA literature [4]. We agree that dividing the ROM in flexion and extension would be an interesting to understand patient’s limitation but that it would be more appropriate for a specific study focused on gait. Regarding thorax and pelvis velocities, iTUG studies sometimes use angular/linear velocities of the pelvis [5] (or waist [6]), and sometime velocities of the thorax [2]. We chose to take both in the analysis and to identify the most relevant in this specific study with PCA.”

Well, many TUG studies investigate the spatiotemporal parameters and, in my opinion, it makes sense. For example, the number of steps, step time, single support length, step length etc. can be very informative in terms of patient agility and potential limping.

How where the kinematics calculated for the different phases? Was it the mean of all steps within the phase for each side or both sides? Furthermore, only the fastest trial was chosen. Why not the mean of the 10 trials? Especially, as you mention the study of Sangeux [1], who did not really investigate the variance of consecutive steps but rather number of trials. They recommend 6 trials for kinematics. Based thereon, the results of the current study would be questionable as well, given that only one TUG trial was analyzed. Given that the gait kinematics most likely deviate from normal around the phase transitions.

[2] did not report differences in spatiotemporal parameters between the TUG (3m) and iTUG (7m), despite the time. A recommendation was not given. Only the conclusion that spatial parameters are less reliable due to the method (accelerometer/gyroscope).

[4] reported ROM probably out of convenience from max. flexion and extension. The included studies, such as, Klausmeier V. (2010) and Mayr E. (2009) report the flexion/extension separately, which makes a lot of sense.

[5] and [6] are accelerometer/smartphone studies and the velocity site is most likely defined by the limited application possibilities.

In conclusion, the variable choice remains inconclusive, and a good rationale is missing. On the one hand, the authors identify the most relevant variables using the PCA but on the other hand, cherry pick and limit their pre-selection to variables which clearly stem and are defined by the convention/limitations of accelerometer or IMU systems.

Two approaches would be conclusive:

1. Use ALL variables previously used in TUG tests to be able to compare results to accelerometer based iTUGs and to identify the most relevant.

2. Use evidence-based variables, which were previously identified to be relevant for THA patients.

- We respectfully disagree with reviewer 2 on this matter, which was interestingly not raised by reviewer 1. As stated in our revised manuscript, we have added a supplementary material describing specifically the choice and computation of the features (S4). The approach suggested by the reviewer (“taking ALL variables previously used in TUG test”) does not seem to be realistic or relevant for the present paper since, as an example, parameters used to assess the upper limbs of patients with Parkinson disease are not applicable for patients with total hip arthroplasty. Moreover, this suggestion would require a specific study dedicated to the choice of variables through a systematic review of literature, which is not within the scope of the present paper. The second approach suggested is to use “evidence-based variables […] relevant for THA patients”. We believe that this is in accordance with the approach we followed, since all the features have specific references describing their previous use in the literature (S4). Since our paper is the first iTUG study with THA patients, not all references are related to THA patients but we chose mainly studies with cohorts of elderly patients which will compare somewhat with our own cohort. We would appreciate your opinion on this issue. We can add more references to S4 but it does not seem realistic for us to start this study from scratch at this point.

Authors: “- Unpaired Student’s t-tests were added to table 1 to assess the differences between groups. However, we do not understand why the test for the weight should be nonparametric. Could the reviewer develop on this point?”

The K-S test is used to test the assumption of normal distribution a t-test. A significant result in the K-S test indicates a non-normal distribution and a violation of the t-test assumption. In this case a non-parametric test is warranted.

Commonly the scientific community assumes that the researchers know and use the appropriate assumption tests for their statistical analysis and usually do not ask to report the assumption test results. The K-S test results can be omitted in the table, but it should be indicated that a non-parametric test is used to compare the weight.

- It seems that there was a misunderstanding with the KS-test. The values reported in Table 1 are two-sample KS test used to compare the groups. When testing for normal distribution with the one-sample KS test, the parameters of both groups are normal, thus, the t-test is appropriate in all cases. Moreover, as recommended by reviewer 1, a Chi-square test was used to check whether the patient and control group had different distribution of sex. The KS-test column was removed from the table and the manuscript modified as follow:

L154: “The difference in sex distribution was tested using a χ2 test. The Kolmogorov–Smirnoff [23] test was used to assess the normality of the age, weight and height distributions among patients and controls. Differences between groups were then tested using unpaired Student’s t-tests.”

Authors: “If the reviewer prefers, we can remove the mention of “pregnancy or breast-feeding patients” from the manuscript.”

If the authors think it is relevant, it can remain in the manuscript.

- This mention was kept in the manuscript.

Reviewer:

• The section “Implementation in clinical settings” might be well-meant, however, has too little relation to the results.

Authors:

- This study was a first step toward the development of clinical tools for the measurements of function. Thus, we believe that this section can be of interest to provide a broader context and expected applications of the study to the reader. However, if the reviewer thinks this is section should be left out we can remove it from the paper.

Reviewer: Since it was not subject of the current investigation, I would exclude this part.

- We would like to insist on this point which was not raised by reviewer 1. This sections seems relevant to us as it provides a broader context to the present study and the direction of future research on this topic. Could the editor decide on whether we should include this section?

Reviewer:

• “hospital length of stay after surgery” hospitalization?

Authors:

- This term is commonly used in medical journals, it represents the time from the day of admission to the day of discharge. See reference [12] in the manuscript: Petis et al. (2016).

-Petis SM, Howard JL, Lanting BA, Somerville LE, Vasarhelyi EM. Perioperative Predictors of Length of Stay After Total Hip Arthroplasty. J Arthroplasty. 2016;31: 1427–1430. doi:10.1016/j.arth.2016.01.005

Reviewer: It remains a clumsy wording and weakens the readability but since it is minor issue the authors may choose to stick to their wording.

- The term “hospital length of stay” has been kept in the manuscript.

Further minor corrections:

- The paper was proofread by a native English speaker: Darren Hart of “publish-or-perish.ch”, a professional proofreader. Your suggestion were implemented before the final proofreading.

Methods:

“The level of statistical significance was controlled with the Holm method (alpha = 0.05)”

Better: To account for multiple testing the Holm’s method was used to adjust the p-values.

- Thank you, this sentence was modified accordingly.

L119: Student t-tests change to Student’s t-test.

- The manuscript was modified accordingly.

P17L235: “although not statistically significant with the Holm.”

Was it significant different without the Holm correction? Either it is significant or not. Either the authors acknowledge the method or not.

- This mention was removed from the manuscript:

L233: “The walking phase showed the greatest mean improvement in function (16%, 95%CI: 8% to 24%, Table 5), although this was not statistically significant.”

P17L235: a similar increase.

- The manuscript was modified accordingly.

P18L269ff: “This feature had the lowest number of patients at the level of the control groups before and 6 months after surgery (8% and 27% respectively).”

Hard to comprehend. Please rephrase.

- The manuscript was modified as follow:

L 273: “Indeed, this feature presented the lowest percentage of patients with a level of function similar to that of the control group, both before and after surgery (8% and 27%, respectively), as well as the greatest number of patients with a positive change in function after surgery (82%).”

P19L281: with a strong

- The manuscript was modified accordingly.

P19L285ff.: “The total time could be a good indicator of speed parameters but do not seem to reflect as well the quality of movement such as the presence of compensatory movements (e.g. high range of thorax obliquity).”

Check for errors, maybe rephrase.

- The manuscript was modified as follow:

L 289: “Total TUG test time could be a good indicator of speed parameters, but it does not seem to reflect comparably the quality of movements, such as the presence of compensatory movements (e.g. high range of thorax obliquity).”

P20L305: “help to identify patients at risks of lowering their level of function after surgery” Appears wrong: implies that patients actively lower their function but it’s something that happens to them.

- The manuscript was modified as follow:

L 308: “Understanding the preoperative factors at the origin of those differences in profile could help to identify the patients at risk of having a lower level of function after surgery and improve patient rehabilitation.”

P20L307: “high variability observed with the analysis of the TUG”…the variability is a result of the analysis and not an observation of the analysis. Please revise the sentence/wording.

- The manuscript was modified as follow:

L 311: “The high variability resulting from the analysis of the TUG test’s phases suggests that it could be used to identify patient-specific levels of function.”

P22L348ff.: “The patient population and control group had a significant difference in term of weight, but a higher weight was reported as a risk factor for hip osteoarthritis [43] which could explain this difference.”

Is not really a limitation, at least not how it is stated. Weight might be a limitation in terms of different kinematics or measurement inaccuracies due to marker placement or skin artifacts.

- From our point of view, the ideal control group would be healthy patients matched in age, sex and weight to have the least differences in term of population characteristics. Cimolin et al. (2019) showed a decreased time of the TUG test for severely obese women (BMI of 41.1 kg/m2) when compared to healthy controls (BMI of 22.8 kg/m2). The difference in BMI is nothing comparable in our study (BMI of 28.8 kg/m2 and 25.2 kg/m2 for patients and controls respectively) but it could be a source of differences in the features and thus, a limitation of the present study. 

P22L360ff:

“The feature “distance chair to start of turn” was chosen with the assumption that a longer distance would imply that the patients turned and sat at the same time, i.e. with an overlapping transition strategy. This feature was continuous, as this study aimed at assessing percentages of differences and change, but, a categorical feature classifying the strategy of the patients could be of interest to understand the patient’s function.”

Why is this a limitation?

- The limitation here is the choice of the feature. It underlines that different features could have been selected to assess the iTUG. Here we gave an example of the fact that two features can represent the same phenomenon: the two different strategies to perform the turn-to-sit task.

---

## [Decision Letter · Decision Letter 2]

9 Jul 2021

Which functional tasks present the largest deficits for patients with total hip arthroplasty before and six months after surgery? A study of the Timed Up-and-Go test phases

PONE-D-20-38214R2

Dear Dr. Gasparutto,

We’re pleased to inform you that your manuscript has been judged scientifically suitable for publication and will be formally accepted for publication once it meets all outstanding technical requirements.

Kind regards,

Peter Andreas Federolf

Academic Editor

PLOS ONE

Additional Editor Comments (optional):

Reviewers' comments:

Reviewer's Responses to Questions

**Comments to the Author**

1. If the authors have adequately addressed your comments raised in a previous round of review and you feel that this manuscript is now acceptable for publication, you may indicate that here to bypass the “Comments to the Author” section, enter your conflict of interest statement in the “Confidential to Editor” section, and submit your "Accept" recommendation.

Reviewer #1: All comments have been addressed

Reviewer #2: All comments have been addressed

2. Is the manuscript technically sound, and do the data support the conclusions?

Reviewer #1: Yes

Reviewer #2: Yes

3. Has the statistical analysis been performed appropriately and rigorously? 

Reviewer #1: Yes

Reviewer #2: Yes

4. Have the authors made all data underlying the findings in their manuscript fully available?

Reviewer #1: Yes

Reviewer #2: Yes

5. Is the manuscript presented in an intelligible fashion and written in standard English?

Reviewer #1: Yes

Reviewer #2: Yes

6. Review Comments to the Author

Reviewer #1: The authors have addressed all my comments. Some of the proof reader’s corrections resulted in uncommon wordings, especially regarding the use of the possessive ‘s (although probably grammatically correct). They’re very minor remarks, so the authors can choose to go back to their original wordings or stay with the current version.

Some examples:

Line 84: The study’s objective (“the objective of the study” is more common)

Line 119: Marker trajectories (instead of markers’ trajectories) is commonly used

Line 121: using a fourth-order Butterwort filter (instead of design)

Line 197: each phase’s duration (“duration of each phase” is more common)

Line 293: in all the test’s phases (“all the phases of the test” is more common)

Reviewer #2: Thank you for implimenting the changes. Even if I disagree in some cases the study adds to the outcome evaluation of THA even if two important planes of motion were neglected.

The paper can be published.

7. PLOS authors have the option to publish the peer review history of their article (what does this mean?). If published, this will include your full peer review and any attached files.

Reviewer #1: No

Reviewer #2: No

---

## [Editor Report · Acceptance letter]

2 Sep 2021

PONE-D-20-38214R2 

Which functional tasks present the largest deficits for patients with total hip arthroplasty before and six months after surgery? A study of the Timed Up-and-Go test phases 

Dear Dr. Gasparutto:

I'm pleased to inform you that your manuscript has been deemed suitable for publication in PLOS ONE. Congratulations! Your manuscript is now with our production department. 

Kind regards, 

on behalf of

Dr. Peter Andreas Federolf 

Academic Editor

PLOS ONE